# Human-specific *ARHGAP11B* induces hallmarks of neocortical expansion in developing ferret neocortex

Nereo Kalebic, Carlotta Gilardi, Mareike Albert, Takashi Namba, Katherine R Long, Milos Kostic, Barbara Langen, Wieland B Huttner*

Max Planck Institute of Molecular Cell Biology and Genetics, Dresden, Germany

**Abstract** The evolutionary increase in size and complexity of the primate neocortex is thought to underlie the higher cognitive abilities of humans. *ARHGAP11B* is a human-specific gene that, based on its expression pattern in fetal human neocortex and progenitor effects in embryonic mouse neocortex, has been proposed to have a key function in the evolutionary expansion of the neocortex. Here, we study the effects of *ARHGAP11B* expression in the developing neocortex of the gyrencephalic ferret. In contrast to its effects in mouse, *ARHGAP11B* markedly increases proliferative basal radial glia, a progenitor cell type thought to be instrumental for neocortical expansion, and results in extension of the neurogenic period and an increase in upper-layer neurons. Consequently, the postnatal ferret neocortex exhibits increased neuron density in the upper cortical layers and expands in both the radial and tangential dimensions. Thus, human-specific *ARHGAP11B* can elicit hallmarks of neocortical expansion in the developing ferret neocortex.
DOI: https://doi.org/10.7554/eLife.41241.001

## Introduction

The expansion of the neocortex during primate evolution is thought to constitute one important basis for the unparalleled cognitive abilities of humans. The size of the neocortex is mainly regulated by the proliferative capacity of neural progenitor cells during cortical development and the length of the neurogenic period (*Azevedo et al., 2009*; *Borrell and Götz, 2014*; *Dehay et al., 2015*; *Kaas, 2013*; *Kalebic et al., 2017*; *Krubitzer, 2007*; *Lui et al., 2011*; *Molnár et al., 2006*; *Rakic, 2009*; *Sousa et al., 2017*; *Wilsch-Bräuninger et al., 2016*).

Two major classes of neural progenitors can be distinguished: apical progenitors (APs), whose cell bodies reside in the ventricular zone (VZ), and basal progenitors (BPs), whose cell bodies reside in the subventricular zone (SVZ). Whereas APs are highly proliferative in the neocortex of all mammalian species studied (*Götz and Huttner, 2005*; *Rakic, 2003a*), BPs are highly proliferative only in species with an expanded neocortex (*Borrell and Götz, 2014*; *Florio and Huttner, 2014*; *Lui et al., 2011*; *Reillo et al., 2011*). Specifically, a subtype of BPs, called basal (or outer) radial glia (bRG), are thought to play a key role in the evolutionary expansion of the neocortex (*Borrell and Götz, 2014*; *Florio and Huttner, 2014*; *Lui et al., 2011*). Importantly, in species with an expanded neocortex, such as primates or the ferret, the SVZ has been shown to be divided into two distinct histological zones: the inner and outer SVZ (ISVZ and OSVZ, respectively) (*Dehay et al., 2015*; *Reillo and Borrell, 2012*; *Smart et al., 2002*). The OSVZ is uniquely important for the evolutionary expansion of the neocortex, as proliferative bRG are particularly abundant in this zone (*Betizeau et al., 2013*; *Fietz et al., 2010*; *Hansen et al., 2010*; *Poluch and Juliano, 2015*; *Reillo and Borrell, 2012*; *Reillo et al., 2011*; *Smart et al., 2002*). Increased proliferative capacity of bRG results in an amplification of BP number and is accompanied by a prolonged phase of production of late-born neurons

*For correspondence:
huttner@mpi-cbg.de

Competing interests: The authors declare that no competing interests exist.

**eLife digest** The human brain owes its characteristic wrinkled appearance to its outer layer, the cerebral cortex. All mammals have a cerebral cortex, but its size varies greatly between species. As the brain evolved, the neocortex, the evolutionarily youngest part of the cerebral cortex, expanded dramatically and so had to fold into wrinkles to fit inside the skull. The human neocortex is roughly three times bigger than that of our closest relatives, the chimpanzees, and helps support advanced cognitive skills such as reasoning and language. But how did the human neocortex become so big?

The answer may lie in genes that are unique to humans, such as *ARHGAP11B*. Introducing *ARHGAP11B* into the neocortex of mouse embryos increases its size and can induce folding. It does this by increasing the number of neural progenitors, the cells that give rise to neurons. But there are two types of neural progenitors in mammalian neocortex: apical and basal. A subtype of the latter – basal radial glia – is thought to drive neocortex growth in human development. Unfortunately, mice have very few basal radial glia. This makes them unsuitable for testing whether *ARHGAP11B* acts via basal radial glia to enlarge the human neocortex.

Kalebic et al. therefore introduced *ARHGAP11B* into ferret embryos in the womb. Ferrets have a larger neocortex than mice and possess more basal radial glia. Unlike in mice, introducing this gene into the ferret neocortex markedly increased the number of basal radial glia. It also extended the time window during which the basal radial glia produced neurons. These changes increased the number of neurons, particularly of a specific subtype found mainly in animals with large neocortex and thought to be involved in human cognition.

Introducing human-specific *ARHGAP11B* into embryonic ferrets thus helped expand the ferret neocortex. This suggests that this gene may have a similar role in human brain development. Further experiments are needed to determine whether ferrets with the *ARHGAP11B* gene, and thus a larger neocortex, have enhanced cognitive abilities. If they do, testing these animals could provide insights into human cognition. The animals could also be used to model human brain diseases and to test potential treatments.

DOI: https://doi.org/10.7554/eLife.41241.002

(*Geschwind and Rakic, 2013*; *Otani et al., 2016*; *Rakic, 2009*). As the mammalian cerebral cortex is generated in an inside-out fashion, these late-born neurons occupy the upper-most layers of the cortex (*Lodato and Arlotta, 2015*; *Molnár et al., 2006*; *Molyneaux et al., 2007*; *Rakic, 1972*; *Rakic, 2009*; *Sidman and Rakic, 1973*). Thus, an increased generation of upper-layer neurons and increased thickness of the upper layers are also hallmarks of an expanded neocortex.

The evolutionary expansion of the neocortex is characteristically accompanied by an increase in the abundance of proliferative bRG, in the length of the neurogenic period, and in the relative proportion of upper-layer neurons within the cortical plate (*Borrell and Götz, 2014*; *Dehay et al., 2015*; *Florio and Huttner, 2014*; *Geschwind and Rakic, 2013*; *Lui et al., 2011*; *Molnár et al., 2006*; *Sousa et al., 2017*; *Wilsch-Bräuninger et al., 2016*). This is most obvious when comparing extant rodents, such as mouse, with primates, such as human. Carnivores, such as ferret, display intermediate features (*Borrell and Reillo, 2012*; *Hutsler et al., 2005*; *Kawasaki, 2014*; *Reillo et al., 2011*). Specifically, ferrets exhibit a gyrified neocortex and, during development, a pronounced OSVZ populated with proliferative bRG (*Barnette et al., 2009*; *Borrell and Reillo, 2012*; *Fietz et al., 2010*; *Kawasaki, 2014*; *Kawasaki et al., 2013*; *Poluch and Juliano, 2015*; *Reillo et al., 2011*; *Sawada and Watanabe, 2012*; *Smart and McSherry, 1986a*; *Smart and McSherry, 1986b* ). In this context, it should be noted that in evolution, the split between the lineages leading to mouse and to human occurred a few million years later than that leading to ferret and human (*Bininda-Emonds et al., 2007*).

In addition to the above-mentioned features associated with neocortex expansion in general, certain specific aspects of human neocortex expansion are thought to involve human-specific genomic changes. Recent transcriptomic studies established that certain previously identified human-specific genes (*Bailey et al., 2002*; *Dennis and Eichler, 2016*) are preferentially expressed in neural progenitor cells and have implicated these genes in human neocortex expansion (*Fiddes et al., 2018*; *Florio et al., 2015*; *Florio et al., 2018*; *Florio et al., 2016*; *Suzuki et al., 2018*). Among these

genes, the one that showed the most specific expression in human bRG compared to neurons was *ARHGAP11B* (*Florio et al., 2015*). *ARHGAP11B* arose in evolution after the split of the human lineage from the chimpanzee lineage, as a product of a partial gene duplication of *ARHGAP11A*, a gene encoding a Rho GTPase activating protein (*Dennis et al., 2017*; *Florio et al., 2015*; *Florio et al., 2016*; *Kagawa et al., 2013*). Forced expression of *ARHGAP11B* in the embryonic mouse neocortex leads to an increase in BP proliferation and pool size (*Florio et al., 2015*). However, as described above, the mouse exhibits only a minute amount of bRG, a cell type thought to be instrumental for neocortex expansion, and the role of ARHGAP11B on the pool size of bRG, therefore, remains elusive. Additionally, the role of ARHGAP11B on the production of upper-layer neurons, another hallmark of the evolutionary expansion of the neocortex, is also unknown. Here, we study the effects of forced expression of *ARHGAP11B* in the developing ferret neocortex, which already exhibits several features of an expanded neocortex, including an abundance of bRG and of upper-layer neurons, and as such is a suitable model organism to address the role of ARHGAP11B in the evolutionary expansion of the neocortex.

## Results

We expressed ARHGAP11B in the ferret neocortex starting at embryonic day 33 (E33), when both the generation of upper-layer neurons and formation of the OSVZ start (*Martínez-Martínez et al., 2016*). Specifically, we performed *in utero* electroporation of ferrets (*Kawasaki et al., 2012*; *Kawasaki et al., 2013*) at E33 with a plasmid encoding *ARHGAP11B* under the constitutive CAG promoter or an empty vector as control. The analyses of electroporated embryos were performed at four different developmental stages: E37, E40/postnatal day (P) 0, 10 and 16 (*Figure 1—figure supplement 1A*). To be able to visualize the electroporated area, we co-electroporated ARHGAP11B-expressing and control plasmids with vectors encoding fluorescent markers. For postnatal studies, to be able to distinguish the electroporated kits, the ARHGAP11B-expressing plasmid was co-electroporated with a GFP-encoding plasmid, and the control vector with an mCherry-encoding plasmid, or *vice versa*. For the sake of simplicity, we refer to both fluorescent markers as Fluorescent Protein (FP) from here onwards, and both FPs are depicted in green color in all figures.

We detected *ARHGAP11B* transcript by RT-qPCR at all the stages analyzed and only in ferret embryos/kits subjected to *ARHGAP11B in utero* electroporation (*Figure 1—figure supplement 1B–D*). Additionally, immunofluorescence at E37 demonstrated the specific presence of the ARHGAP11B protein in neural progenitors of such embryos (*Figure 1—figure supplement 1E,F* and see Materials and methods for details).

### ARHGAP11B increases the abundance of BPs in the developing ferret neocortex

We first examined the ability of ARHGAP11B to increase BP abundance in ferret. To this end, we immunostained E40/P0 ferret neocortex for PCNA, a marker of cycling cells, in order to identify progenitor cells (*Figure 1A* and *Figure 1—figure supplement 2A*). We observed an increase in the proportion of PCNA+ FP+ cells in OSVZ of the ARHGAP11B-expressing embryos compared to control (*Figure 1B*). The abundance of PCNA+ FP+ cells was increased in both ISVZ and OSVZ, but this increase was particularly strong in the OSVZ (*Figure 1—figure supplement 2B*). Of note, we did not detect any increase in the abundance of FP– progenitor cells in the SVZ, suggesting that ARHGAP11B does not promote any non-cell-autonomous effects (*Figure 1—figure supplement 2C*).

We next immunostained the E40/P0 ferret neocortex for phospho-vimentin (PhVim), a marker of mitotic cells (*Figure 1C*). Our analysis revealed no effect of ARHGAP11B expression on apical mitoses (*Figure 1D*, left-most) and on basal mitoses in the abventricular VZ (*Figure 1D*, second column from left) compared to control. In contrast, a 3-fold increase in the abundance of basal mitotic cells in the SVZ was detected (*Figure 1D*, sum of ISVZ and OSVZ). This increase was observed for the ISVZ (2-fold, *Figure 1D*, second column from right), but was especially prominent for the OSVZ (5-fold, *Figure 1D*, right-most column). A comparably large increase (5-fold) was detected when examining mitotic bRG, that is, PhVim+ BPs exhibiting a PhVim+ process in the OSVZ (*Figure 1E,F*). bRG accounted for ≈50% of all BPs upon ARHGAP11B expression, and their relative proportion was not significantly changed compared to control or non-electroporated regions (*Figure 1—figure supplement 2D*). Of note, this strong increase in FP+ basal mitoses was not accompanied by any change in

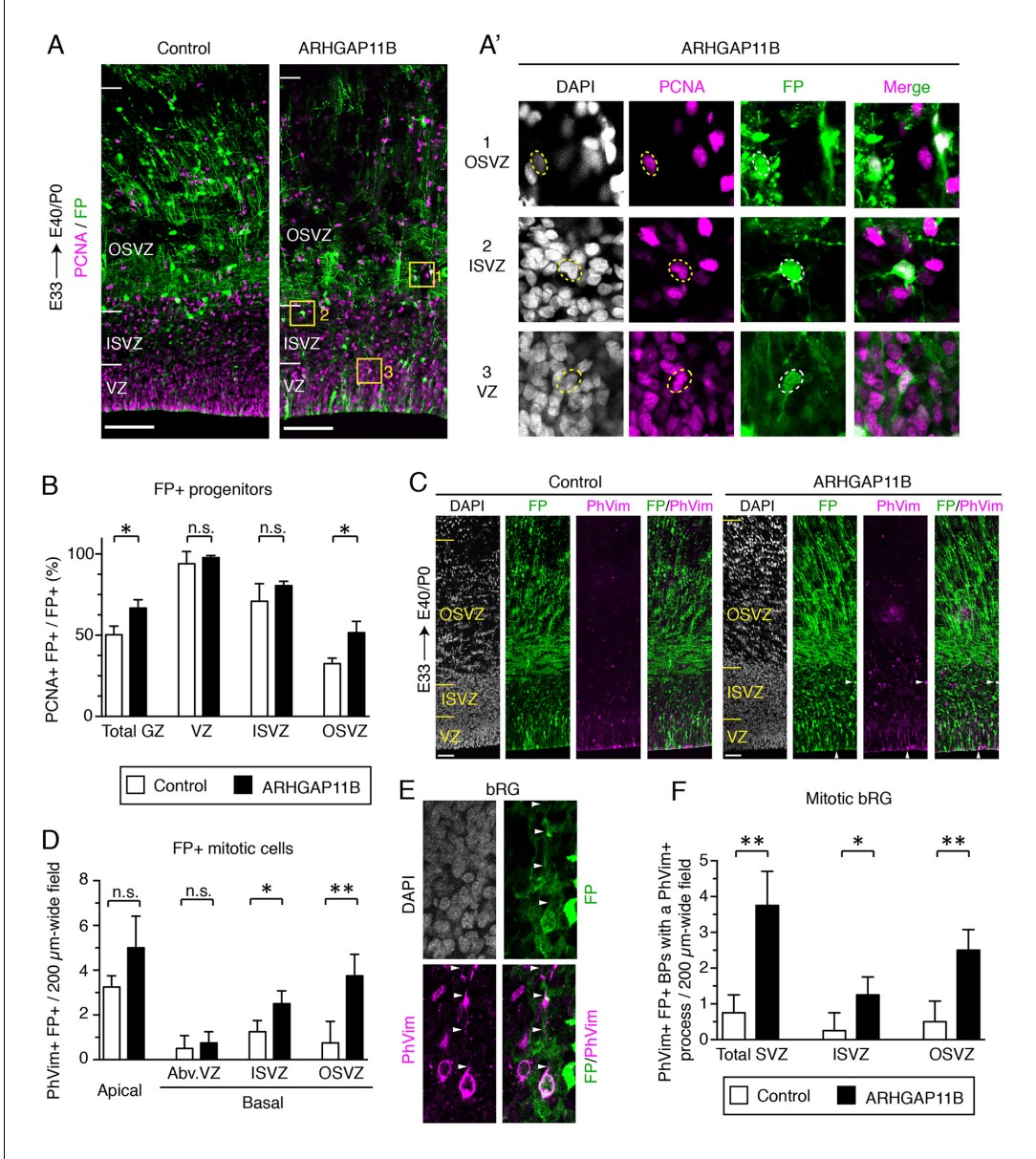

**Figure 1.** ARHGAP11B increases the abundance of BPs in the developing ferret neocortex. Ferret E33 neocortex was electroporated *in utero* with a plasmid encoding a fluorescent protein (FP) together with either a plasmid encoding ARHGAP11B or empty vector (Control), followed by analysis at E40/P0. (**A**) Double immunofluorescence for FP (green) and PCNA (magenta) (for the images of the single channels and DAPI staining, see *Figure 1—figure supplement 2A*). Images are single optical sections. Scale bars, 100 μm. Boxes (50 × 50 μm) indicate FP+ BPs in the OSVZ (1, top), ISVZ (2, middle) and VZ (3, bottom), shown at higher magnification in (**A**). (**A**) Dashed lines indicate a cell body contour. (**B**) Percentage of FP+ cells in the germinal zones (GZ total) and in the VZ, ISVZ and OSVZ that are PCNA+ upon control (white) and ARHGAP11B (black) electroporations. Data are the mean of 3 experiments. Error bars indicate SD; *, p <0.05; n.s., not statistically significant; Student's *t*-test. (**C**) Double immunofluorescence for FP (green) and phospho-vimentin (PhVim, magenta), combined with DAPI staining (white). Images are single optical sections. Scale bars, 50 μm. Vertical arrowheads, apical mitosis; horizontal arrowheads, basal mitosis. (**D**) Quantification of FP+ mitotic cells, as revealed by PhVim immunofluorescence, in a 200 μm-wide field of the cortical wall, upon control (white) and ARHGAP11B (black) electroporations. Apical, mitoses lining the ventricular surface; basal, mitoses away from the ventricle (Abv.VZ, abventricular VZ; ISVZ; OSVZ). Data are the mean of 4 experiments. Error bars indicate SD; **, p <0.01; *, p <0.05; n.s., not statistically significant; Student's *t*-test. (**E**) Mitotic bRG (single optical sections). Double immunofluorescence for FP (green) and phospho-vimentin (PhVim, magenta), combined with DAPI staining (white), upon electroporation of the plasmid encoding FP together with the plasmid encoding ARHGAP11B. Arrowheads, PhVim+ basal process of the mitotic bRG. Images are oriented with the apical side facing down and are 25 μm wide. (**F**) Quantification of mitotic bRG (FP+ PhVim+ cell bodies in the SVZ that contain a PhVim+ process), in a 200 μm-wide field of total SVZ (left), ISVZ (middle) and OSVZ (right), upon control (white) and ARHGAP11B (black) electroporations. Data are the mean of 4 experiments. Error bars indicate SD; **, p <0.01; *, p <0.05; Student's *t*-test.

*Figure 1 continued on next page*

*Figure 1 continued*

DOI: https://doi.org/10.7554/eLife.41241.003

The following figure supplements are available for figure 1:

**Figure supplement 1.** Forced expression of ARHGAP11B in the developing ferret neocortex.

DOI: https://doi.org/10.7554/eLife.41241.004

**Figure supplement 2.** ARHGAP11B increases the abundance of BPs in the developing ferret neocortex.

DOI: https://doi.org/10.7554/eLife.41241.005

FP– mitotic cells (*Figure 1—figure supplement 2E*) nor by a change in thickness of the ferret germinal zones (*Figure 1—figure supplement 2F*). Taken together, these data indicate that ARHGAP11B markedly increases the abundance of BP, in particular bRG, when expressed in the embryonic ferret neocortex.

## ARHGAP11B increases the proportion of Sox2-positive bRG that are Tbr2-negative

We next analyzed the ARHGAP11B-increased bRG in more detail. Proliferative neural progenitors, in particular apical radial glia (aRG) and bRG, characteristically express the transcription factor Sox2 (*Pollen et al., 2015*). We therefore immunostained E40/P0 ferret neocortex for Sox2 (*Figure 2A*) and detected a 40% increase in the proportion of Sox2+ FP+ cells in the germinal zones (GZs) (*Figure 2B*). This increase was exclusively due to an increase in BPs, as we observed a doubling of the proportion of Sox2+ FP+ cells in both the ISVZ and OSVZ, but no increase in the VZ (*Figure 2C*), upon ARHGAP11B expression. These data in turn are consistent with the effects of ARHGAP11B described above (*Figure 1—figure supplement 2B*).

We then analyzed the FP+ cells for the expression of the transcription factor Tbr2 (*Figure 2A*), a marker of certain BPs (*Englund et al., 2005*). In embryonic mouse neocortex, Tbr2 is not only expressed in the predominant type of BP, the basal intermediate progenitors (bIPs) which are known to be neurogenic (*Haubensak et al., 2004*; *Miyata et al., 2004*; *Noctor et al., 2004*), but also (in contrast to an earlier report (*Wang et al., 2011*)) in the vast majority of mitotic bRG (*Florio et al., 2015*), which exhibit a low proliferative capacity (*Wang et al., 2011*). In contrast, human bRG, which exhibit a high proliferative capacity (*Hansen et al., 2010*; *LaMonica et al., 2013*), largely lack Tbr2 expression (*Fietz et al., 2010*; *Hansen et al., 2010*). Extrapolating from these data on mouse and human BPs to the bRG in embryonic ferret neocortex, many of which express Tbr2 (*Reillo et al., 2011*), it appears justified to assume that a portion of the latter bRG may be neurogenic and exhibit a reduced proliferative capacity. Consistent with this assumption and with the effects of ARHGAP11B expression on neural progenitors in the developing ferret neocortex described so far, we observed, upon expression of ARHGAP11B for 7 days, a decrease in the proportion of Tbr2+ FP+ progenitors in all GZs, which was statistically significant for the VZ (*Figure 2D,E*).

In order to potentially obtain further cues as to the proliferative capacity of the ARHGAP11B-increased bRG in the developing ferret neocortex, we focused our attention on progenitor cells that (i) exhibited a radial morphology, (ii) expressed Sox2, but (iii) lacked Tbr2 expression (*Figure 2F*). Upon ARHGAP11B expression, we observed, in the sum of the GZs, a 20% increase in the proportion of radial Sox2+ cells that were Tbr2– (*Figure 2G*). This was largely due to an increase in the proportion of these cells in the OSVZ (*Figure 2H*), where more than 90% were Tbr2–. Considering that the ISVZ is the GZ with the highest amount of Tbr2+ BPs (*Figure 2A*), we examined the relative proportions of Sox2+ Trb2– and Sox2+ Tbr2+ BPs in the ISVZ and did not observe any significant difference in these proportions between control and ARHGAP11B expression (*Figure 2—figure supplement 1*).

These findings are consistent with the notion that the ARHGAP11B-increased radial Sox2+ Tbr2– cells in the OSVZ are bRG. Studies in fetal human neocortex have established that such cells in the OSVZ are highly proliferative (*Hansen et al., 2010* and *LaMonica et al., 2013*; for reviews see *Florio and Huttner, 2014* and *Lui et al., 2011*). Hence, our finding that expression of ARHGAP11B in developing ferret neocortex results in a marked increase in the proportion of these cells suggests that this human-specific gene is sufficient to promote, in a gyrencephalic carnivore, the generation of bRG with putatively increased proliferative capacity.

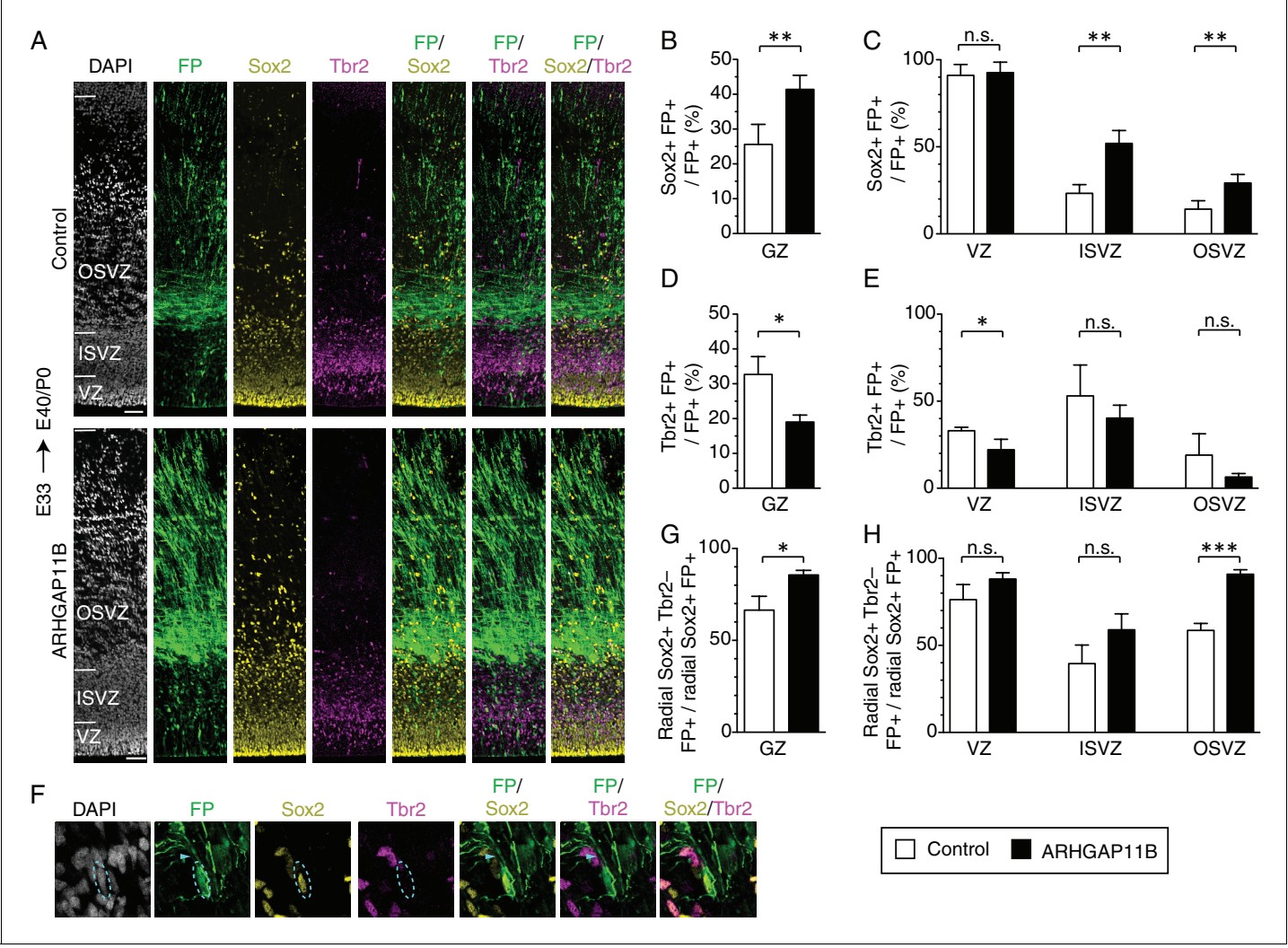

**Figure 2.** ARHGAP11B increases the proportion of Sox2-positive bRG that are Tbr2-negative. Ferret E33 neocortex was electroporated *in utero* with a plasmid encoding FP together with either a plasmid encoding ARHGAP11B or empty vector (Control), followed by triple immunofluorescence for FP (green), Sox2 (yellow) and Tbr2 (magenta), combined with DAPI staining (white), at E40/P0. (**A**) Overview of the electroporated areas (single optical sections). Scale bars, 50 μm. (**B, C**) Percentage of FP+ cells in the germinal zones (B, GZ) and in the VZ (C, left), ISVZ (C, center) and OSVZ (C, right) that are Sox2+ upon control (white) and ARHGAP11B (black) electroporations. (**D, E**) Percentage of FP+ cells in the germinal zones (D, GZ) and in the VZ (E, left), ISVZ (E, center) and OSVZ (E, right) that are Tbr2+ upon control (white) and ARHGAP11B (black) electroporations. (**F**) Proliferative bRG (Sox2 + Tbr2– cell in the SVZ exhibiting radial morphology, single optical sections). Triple immunofluorescence for FP (green), Sox2 (yellow) and Tbr2 (magenta), combined with DAPI staining (white), upon electroporation of the plasmid encoding FP together with the plasmid encoding ARHGAP11B. Dashed lines, cell body; arrowheads, radial process. Images are oriented with the apical side facing down and are 25 μm wide. (**G, H**) Percentage of Sox2+ FP+ cells exhibiting radial morphology in the germinal zones (G, GZ) and in the VZ (H, left), ISVZ (H, center) and OSVZ (H, right) that are Tbr2– upon control (white) and ARHGAP11B (black) electroporations. (**B–E, G, H**) Data are the mean of 4 experiments. Error bars indicate SD; ***, p <0.001; **, p <0.01; *, p <0.05; n.s., not statistically significant; Student's *t*-test.

DOI: https://doi.org/10.7554/eLife.41241.006

The following figure supplement is available for figure 2:

**Figure supplement 1.** ARHGAP11B expression does not affect the proportion of Sox2+ Tbr2+ vs. Sox2+ Tbr2– cells in the ISVZ.
DOI: https://doi.org/10.7554/eLife.41241.007

## ARHGAP11B expression in developing ferret neocortex results in an extended neurogenic period

We investigated the potential consequences of the ARHGAP11B-elicited increase in the abundance of BPs, notably of Sox2+ Tbr2– bRG, for neurogenesis in the developing ferret neocortex. To this

end, we immunostained E40/P0 ferret neocortex for Tbr1, a transcription factor which is a marker of deep-layer neurons (*Kolk et al., 2006*) (*Figure 3—figure supplement 1A*), and for Satb2, a transcriptional regulator which is expressed in neurons that establish callosal projections and that are highly enriched in the upper layers of the cortical plate (CP) (*Alcamo et al., 2008*; *Britanova et al., 2008*) (*Figure 3A*). The vast majority (>90%) of the FP+ neurons in the CP of both control and ARHGAP11B-expressing ferret neocortex were found to be Satb2+ (*Figure 3—figure supplement 1B*). This high percentage is consistent with our experimental approach in which we targeted the embryonic ferret neural progenitors by *in utero* electroporation at E33, that is the time when the generation of the upper-layer neurons starts (*Jackson et al., 1989*; *Martínez-Martínez et al., 2016*).

Analysis of the distribution of Satb2+ FP+ neurons at E40/P0 between the CP on the one hand side, and the GZs plus the intermediate zone (IZ) on the other hand side, revealed that around 60% of the neurons had reached the CP in both control and ARHGAP11B-expressing ferret neocortex (*Figure 3B* left). We next examined electroporated ferret neocortex at P10 (*Figure 3—figure supplement 2*; analysis confined to gyri), which is the stage when neuron production is completed and neuron migration is terminating in the motor and somatosensory areas (*Jackson et al., 1989*; *Smart and McSherry, 1986a*; *Smart and McSherry, 1986b*). Consistent with this, our analysis of the control brains revealed that more than 90% of the Satb2+ FP+ neurons had reached the CP (*Figure 3B* middle). In contrast, only 70% of the Satb2+ FP+ neurons were found in the CP of the ARHGAP11B-expressing neocortex (*Figure 3B* middle). However, at P16, nearly all Satb2+ FP+ neurons were found in the CP in both control and ARHGAP11B-expressing neocortex (*Figure 3B* right; again, analysis confined to gyri). These data suggested that upon ARHGAP11B expression, either neurons migrate more slowly to the CP, or the neurogenic period is extended.

To explore the latter scenario, we injected EdU into P5 ferret kits (i.e. 12 days after electroporation), which is the stage when neuronal progenitors undergo their very last neuron-generating cell divisions in the motor and somatosensory areas of the neocortex (*Jackson et al., 1989*; *Smart and McSherry, 1986a*; *Smart and McSherry, 1986b*). Analysis 11 days after EdU injection, at P16 (*Figure 3C*), revealed a 4-fold increase in the proportion of FP+ cells that were EdU+, in neocortical gyri of ARHGAP11B-expressing kits compared to control (*Figure 3D*). This was consistent with a prolonged, and hence increased, production of cells in ARHGAP11B-expressing kits, which in turn would be in line with the above described finding that ARHGAP11B increases the abundance of proliferative bRG. Importantly, among the EdU+ FP+ cells of the ARHGAP11B-expressing neocortex, 20% were Satb2+ neurons (*Figure 3C' and E*). In contrast, we did not detect a single Satb2+ EdU+ FP+ neuron in any of the control neocortices (*Figure 3E*). Collectively, these data indicate that neurogenesis in ARHGAP11B-expressing ferret neocortex continues longer than in control neocortex.

## ARHGAP11B expression in developing ferret neocortex results in a greater abundance of upper-layer neurons

In light of the extension of the neurogenic period upon ARHGAP11B expression, we examined a potential increase in the abundance of the last-born neurons, that is, the upper-layer neurons. To this end, we first performed Nissl staining of the P16 ferret neocortex to visualize all neurons and the various layers of the CP, and immunostaining for Satb2, which is expressed in the majority of upper-layer neurons (*Alcamo et al., 2008*; *Britanova et al., 2008*; *Lodato and Arlotta, 2015*) (*Figure 4A*). These analyses, carried out on gyri, revealed (i) an increase in the abundance of FP+ cells in the CP (*Figure 4—figure supplement 1A*), and (ii) an alteration in the distribution of the FP+ cells between layers II-VI of the CP, with a greater proportion of cells in layers II-IV (*Figure 4—figure supplement 1B*), in the ARHGAP11B-expressing neocortex compared to control. A similar abundance increase (*Figure 4—figure supplement 1C*) and altered distribution (*Figure 4B*) were observed for Satb2+ FP+ neurons. Furthermore, the proportion of FP+ cells in the CP that were Satb2+ neurons was increased upon ARHGAP11B expression (*Figure 4C*).

We sought to corroborate these data by immunostaining the P16 ferret neocortex for Brn2 (Pou3f2, *Figure 4—figure supplement 2A*), a transcription factor considered to be a marker of layer II and III neurons (*Dominguez et al., 2013*; *McEvilly et al., 2002*; *Sugitani et al., 2002*). Quantification of Brn2+ FP+ cells in layers II and III of gyri revealed a marked increase in the abundance of these cells (*Figure 4—figure supplement 2B*) and in the proportion of FP+ cells that were Brn2+ neurons (*Figure 4D*), in ARHGAP11B-expressing kits compared to control. Of note, the effects of

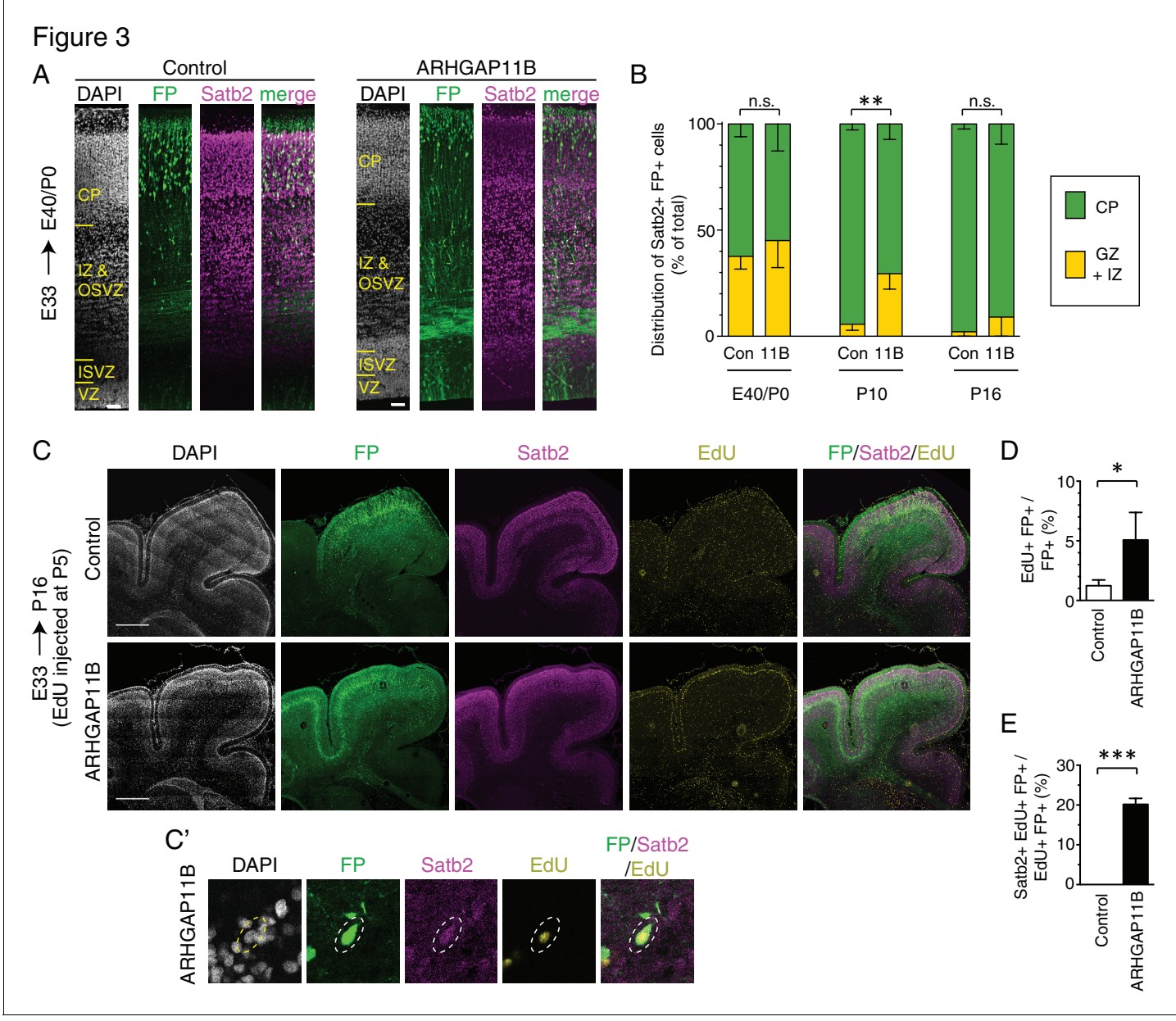

**Figure 3.** ARHGAP11B expression in developing ferret neocortex results in an extended neurogenic period. Ferret E33 neocortex was electroporated *in utero* with a plasmid encoding FP, together with either a plasmid encoding ARHGAP11B or empty vector (Control), followed by analysis at E40/P0 (A, B left), P10 (B center) and P16 (B right, (C–E). (A) Double immunofluorescence for FP (green) and Satb2 (magenta), combined with DAPI staining (white), of the E40/P0 ferret neocortex. The immunofluorescence of the same cryosection for Tbr1 is shown in *Figure 3—figure supplement 1A*. Images are single optical sections. Scale bars, 50 μm. (B) Distribution of Satb2+ FP+ neurons at E40/P0 (left), P10 (center) and P16 (right), between the cortical plate (CP, green) and germinal zones plus intermediate zone (GZ + IZ, yellow), upon control (Con, left) and ARHGAP11B (11B, right) electroporations. Data are the mean of 3 (P0 and P10) or 4 (P16) experiments. Error bars indicate SD; **, p <0.01; n.s., not statistically significant; two-way ANOVA with Bonferroni post-hoc tests (P10, Control CP vs. ARHGAP11B CP, p =0.0015). (C) Triple (immuno)fluorescence for FP (green), Satb2 (magenta) and EdU (yellow), combined with DAPI staining (white), of the P16 ferret neocortex, upon EdU injection at P5. Images are single optical sections. Scale bars, 1 mm. (C') Higher magnification of a FP+ Satb2+ EdU+ neuron upon electroporation of the plasmid encoding FP together with the plasmid encoding ARHGAP11B. Dashed lines, cell body. Images (single optical sections) are oriented with the apical side facing down and are 50 μm wide. (D) Percentage of FP+ cells that are EdU+ upon control (white) and ARHGAP11B (black) electroporations. (E) Percentage of EdU+ FP+ cells that are Satb2 + upon control (white) and ARHGAP11B (black) electroporations. (D, E) Data are the mean of 3 experiments. Error bars indicate SD; ***, p <0.001; *, p <0.05; Student's *t*-test.

DOI: https://doi.org/10.7554/eLife.41241.008

The following figure supplements are available for figure 3:

*Figure 3 continued on next page*

*Figure 3 continued*

**Figure supplement 1.** Almost all neurons generated from ARHGAP11B-expressing progenitors are Satb2+.
DOI: https://doi.org/10.7554/eLife.41241.009
**Figure supplement 2.** FP and Satb2 immunostaining patterns of control and ARHGAP11B-expressing developing ferret neocortex at P10.
DOI: https://doi.org/10.7554/eLife.41241.010

ARHGAP11B expression on the Brn2+ neurons were of a greater degree than those on the Satb2 + neurons (compare *Figure 4—figure supplement 2B* with *Figure 4—figure supplement 1C*; *Figure 4*, panel D with panel C), in line with the notion that Brn2 exhibits a greater specificity for layer II and III neurons than Satb2 (*Dominguez et al., 2013*; *Sugitani et al., 2002*). Taken together, our data indicate that the extension of the neurogenic period upon ARHGAP11B expression in the developing ferret neocortex is accompanied by an increase in the abundance of layer II and III neurons.

## ARHGAP11B expression in developing ferret neocortex leads to its radial and tangential expansion

Finally, we explored whether the ARHGAP11B-elicited increase in the abundance of BPs, notably of proliferative bRG, resulting in an extended neurogenic period and in an increased abundance of upper-layer neurons, had any consequences for the size and morphology of the ferret neocortex. To this end, we analyzed the neocortex at P16 (*Figure 4—figure supplement 3A–C*), the developmental stage when cortical folds have already formed in the ferret (*Barnette et al., 2009*; *Fernández et al., 2016*; *Matsumoto et al., 2017*; *Sawada and Watanabe, 2012*; *Shinmyo et al., 2017*), and quantified a set of morphological parameters (*Figure 4—figure supplement 3D*), each on two coronal sections located slightly rostrally (*Figure 4—figure supplement 3B and C*, position 1) and slightly caudally (*Figure 4—figure supplement 3B and D*, position 2), respectively, to a middle position along the rostro-caudal axis. This revealed no significant differences between control and ARHGAP11B-expressing kits with regard to brain mass (*Figure 4—figure supplement 3E*) and gross neocortex morphology, including the gyrification index of the electroporated area (referred to as local GI), the size of the posterior sigmoid gyrus, lateral gyrus and coronal gyrus, and the depth and thickness of the cruciate sulcus, suprasylvian sulcus and lateral sulcus (*Figure 4—figure supplement 3F–I*; for the gyri and sulci positions, see *Figure 4—figure supplement 3B–D*).

However, we did detect an increase, upon ARHGAP11B expression, in the thickness of the three gyri analyzed (*Figure 4E*). We therefore investigated the underlying cause of this thickness increase and, considering the increased abundance of upper-layer neurons upon ARHGAP11B expression (*Figure 4—figure supplement 1C*, *Figure 4—figure supplement 2B*), measured the thickness of CP layers II-IV in the gyri of control and ARHGAP11B-expressing ferret neocortex at P16. This revealed a 100 µm increase in the thickness of these upper layers in ARHGAP11B-expressing neocortex compared to control (*Figure 4F*, note also the thicker layer II in *Figure 4A* and another example in *Figure 4—figure supplement 5G*). Hence, ARHGAP11B is sufficient to promote neocortex expansion in ferret in the radial dimension.

We then explored whether ARHGAP11B would also be able to promote neocortex expansion in ferret in the tangential dimension. Here, we exploited the previous findings that during the first week of the ferret postnatal neocortex development, migrating neurons change their migration mode from radial to tangential, which results in an increased lateral dispersion of the late-born neurons (*Gertz and Kriegstein, 2015*; *Reillo et al., 2011*; *Smart and McSherry, 1986a*; *Smart and McSherry, 1986b*). In line with this, the abundance of FP+ cells per 200 µm-wide field of cortical wall decreased about 3-fold from E40/P0 to P16 (*Figure 4—figure supplement 4A*). Yet, the increase in FP+ cells per field of cortical wall due to ARHGAP11B expression was not only observed at P16, confirming the above described analysis of FP+ cell numbers in the CP (*Figure 4—figure supplement 1A*), but was already detected at E40/P0 (*Figure 4—figure supplement 4A*). The decrease in FP+ cell abundance per 200 µm-wide field of cortical wall from E40/P0 to P16 (*Figure 4—figure supplement 4A*) was not accompanied by an increase in cell death (*Figure 4—figure supplement 4B–G*). Rather, this decrease reflected an increase in the lateral spread of the progeny of the targeted cells, as evidenced by the ARHGAP11B-elicited increase in the lateral length of the area harbouring FP+ cells that was observed at P16 (*Figure 4G*, *Figure 4—figure supplement 5A*

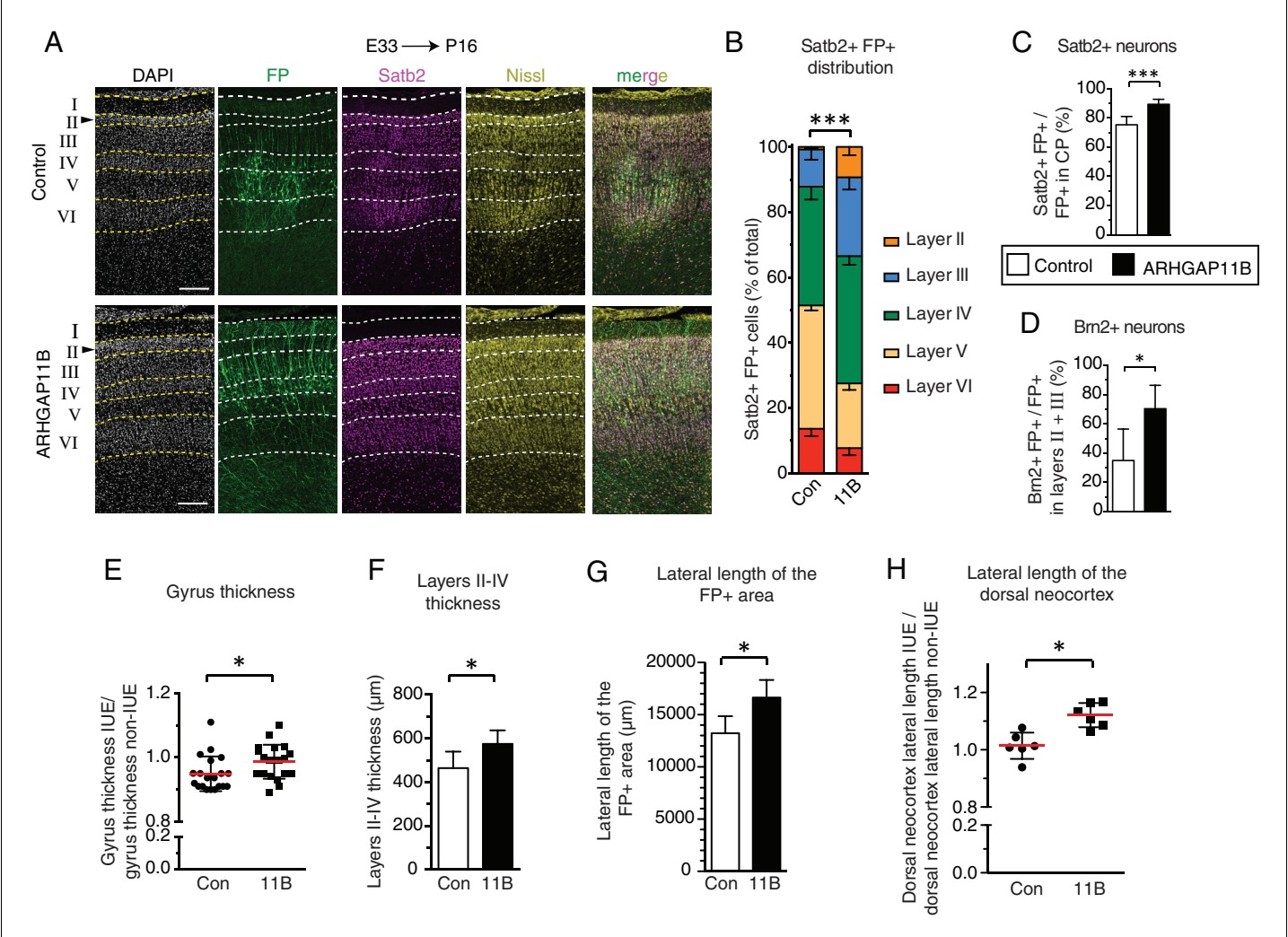

**Figure 4.** ARHGAP11B expression results in a greater abundance of upper-layer neurons and expansion of the developing ferret neocortex. Ferret E33 neocortex was electroporated *in utero* with a plasmid encoding FP, together with either a plasmid encoding ARHGAP11B or empty vector (Control), followed by analysis at P16. (**A**) Double immunofluorescence for FP (green) and Satb2 (magenta), combined with DAPI (white) and Nissl (yellow) staining, of the CP (single optical sections). Neuronal layers are marked on the left. Arrowheads, increased thickness of layer II upon ARHGAP11B expression. Scale bars, 200 μm. (**B**) Distribution of Satb2+ FP+ neurons between the neuronal layers upon control (Con, left) and ARHGAP11B (11B, right) electroporations. Data are the mean of 6 experiments. Error bars indicate SD; ***, p <0.001; two-way ANOVA with Bonferroni post-hoc tests (Layer V, Control vs. ARHGAP11B, p <0.0001; Layer III, Control vs. ARHGAP11B, p =0.0073) (**C**) Percentage of FP+ cells in the CP that are Satb2+, upon control (white) and ARHGAP11B (black) electroporations. Data are the mean of 6 experiments. Error bars indicate SD; ***, p <0.001; Student's *t*-test. (**D**) Percentage of FP+ cells in layers II + III that are Brn2+, upon control (white) and ARHGAP11B (black) electroporations. Data are the mean of 5 experiments. Error bars indicate SD; *, p <0.05; Student's *t*-test. (**E**) Quantification of the gyrus thickness of control (Con) and ARHGAP11B-expressing (11B) ferret neocortex. Measurements were performed as described in *Figure 4—figure supplement 3*. All data are expressed as ratio between electroporated hemisphere (IUE) and non-electroporated contralateral hemisphere (non-IUE). Data are the mean (red lines) of 20 gyri per condition from six neocortices per condition. Error bars indicate SD; *, p <0.05; Student's *t*-test. (**F**) Quantification of layers II-IV thickness, upon control (Con, white) and ARHGAP11B (11B, black) electroporations. Data are the mean of 6 experiments. Error bars indicate SD; *, p <0.05; Student's *t*-test. (**G**) Quantification of the lateral length of the entire areas harbouring FP+ cells, measured as depicted in *Figure 4—figure supplement 5A* top, upon control (Con, white) and ARHGAP11B (11B, black) electroporations. Data are the mean of 6 experiments. Error bars indicate SD; **, p <0.01; Student's *t*-test. (**H**) Quantification of the lateral length of the dorsal neocortex, measured as depicted in *Figure 4—figure supplement 5A* bottom, upon control (Con) and ARHGAP11B (11B) electroporations. All data are expressed as ratio between electroporated hemisphere (IUE) and non-electroporated contralateral hemisphere (non-IUE). Data are the mean of 6 experiments. Error bars indicate SD; *, p <0.05; Student's *t*-test.

DOI: https://doi.org/10.7554/eLife.41241.011

The following figure supplements are available for figure 4:

**Figure supplement 1.** ARHGAP11B expression results in a greater abundance of FP+ cells and FP+ Satb2+ neurons in the CP.

*Figure 4 continued on next page*

*Figure 4 continued*

DOI: https://doi.org/10.7554/eLife.41241.012

**Figure supplement 2.** ARHGAP11B expression results in a greater abundance of FP+ Brn2+ neurons in cortical layers II and III.

DOI: https://doi.org/10.7554/eLife.41241.013

**Figure supplement 3.** ARHGAP11B expression in developing ferret neocortex does not increase neocortical folding.

DOI: https://doi.org/10.7554/eLife.41241.014

**Figure supplement 4.** ARHGAP11B expression in developing ferret neocortex does not lead to increased cell death.

DOI: https://doi.org/10.7554/eLife.41241.015

**Figure supplement 5.** ARHGAP11B expression in developing ferret neocortex leads to its tangential expansion and an increase in cell density.

DOI: https://doi.org/10.7554/eLife.41241.016

**Figure supplement 6.** ARHGAP11B induces the appearance of astrocytes from the targeted progenitors in the developing ferret neocortex.

DOI: https://doi.org/10.7554/eLife.41241.017

top, B). Of note, the increase in the lateral spread of FP+ cells persisted along the rostro-caudal axis of the FP+ region of the neocortex (*Figure 4—figure supplement 5C*) and did not reflect a difference in the electroporation efficiency, as the lateral length of the FP+ area on both the basal and apical side was not different between control and ARHGAP11B-expressing E40/P0 ferret neocortex (*Figure 4—figure supplement 5D,E*).

Importantly, the total lateral length of the dorsal neocortex was also increased upon ARHGAP11B expression (*Figure 4H*, *Figure 4—figure supplement 5A* bottom). This increase was observed at different positions along the rostro-caudal axis (*Figure 4—figure supplement 5F*). These data further imply that the increase in the lateral spread (*Figure 4G*, *Figure 4—figure supplement 5A* top, B) did not occur at the expense of the other, FP–, areas. We conclude that ARHGAP11B is sufficient to promote neocortex expansion in ferret also in the tangential dimension.

Given that the ARHGAP11B-elicited increase in the lateral length of the FP+ area ($\approx 25\%$, *Figure 4G*) was greater than the ARHGAP11B-elicited tangential expansion ($\approx 10\%$, *Figure 4H*), we explored whether this discrepancy could be resolved by ARHGAP11B expression increasing cell density, specifically in the upper layers of the CP where most of the ARHGAP11B-increased neurons resided. Indeed, our analysis revealed an increase in total cell density (revealed by DAPI staining) (*Figure 4—figure supplement 5G,H*) and in neuronal density (revealed by Satb2 immunostaining, *Figure 4—figure supplement 5G,I*) in layers III and IV at P16, but not in layer II, which in general exhibited the highest cell density. Taken together, these data indicate that the ARHGAP11B-induced increase in BP pool size led to an increase in upper-layer neurons that in turn resulted in (i) a thickening of the upper layers, (ii) a higher cell density in those upper layers that were able to accommodate additional neurons, (iii) a greater lateral spread of upper-layer neurons, and (iv) consequently, a greater lateral length of the dorsal neocortex.

## ARHGAP11B induces the appearance of astrocytes from the targeted progenitors in the developing ferret neocortex

Neocortex expansion involves not only an increase in the number of neurons, but also of glial cells. Although we on purpose performed *in utero* electroporation of the ferret neocortex at E33 in order to target OSVZ progenitors generating upper-layer neurons (*Jackson et al., 1989*; *Martínez-Martínez et al., 2016*), and therefore would expect only a small proportion of the FP+ cells to be glial cells, we nonetheless examined the possible occurrence of GFAP+ and S100ß+ cells (as indicators of an astrocytic lineage (*Reillo et al., 2011*; *Voigt, 1989*)) and of Olig2+ cells (as an indicator of an oligodendrocytic lineage (*Mo and Zecevic, 2009*; *Voigt, 1989*)) among the FP+ progeny of the targeted cells. An additional reason to examine astrocytes was that radial glial progenitor cells at later stages of human, monkey and ferret neocortex development are known to differentiate into astrocytic progenitors which then give rise to post-mitotic astrocytic cells (*Rakic, 1978*; *Rakic, 2003b*; *Reillo et al., 2011*; *Voigt, 1989*).

Upon immunostaining of the P16 ferret neocortex for GFAP (*Figure 4—figure supplement 5A,B*) and Olig2 (*Figure 4—figure supplement 5E*), no FP+ cells in the CP of control neocortex that were GFAP+ or Olig2+ could be detected. In contrast, while we still did not detect any Olig2+ FP+ cells in the CP of ARHGAP11B-expressing neocortex, about 3% of the CP FP+ cells were GFAP+ (*Figure 4—figure supplement 5C*). Given that ARHGAP11B expression increases the pool size of FP

+ cells (*Figure 4—figure supplement 1A*, *Figure 4—figure supplement 4A*), this translates into an ARHGAP11B-induced appearance of GFAP+ FP+ cells in the CP. All of these GFAP+ FP+ cells were Ki67– (data not shown) and hence post-mitotic astrocytes. To corroborate these data, we performed immunostaining of the P16 ferret neocortex for another astrocyte marker, S100ß (Figure 4–figure supplement 5D) (*López-Hidalgo et al., 2016*), and readily identified FP+ S100ß+ astrocytes upon ARHGAP11B expression. These data are consistent with at least two scenarios related to the increased abundance of proliferative bRG upon ARHGAP11B expression. Either this increase results not only in an extension of the neurogenic period but also in the initiation of astrogenesis. Or the greater bRG pool size gives rise to a detectable level of astrocytic cells once the bRG differentiate along the astrocytic lineage (*Rakic, 2003b*; *Reillo et al., 2011*; *Voigt, 1989*).

## Discussion

We have shown that expression of a single human-specific gene, *ARHGAP11B*, which has been implicated in the evolutionary expansion of the human neocortex (*Florio et al., 2015*; *Florio et al., 2016*), is sufficient to elicit features in the developing ferret neocortex that are characteristically associated with a further expanded neocortex such as human. These features include (i) an increase in the pool size of proliferative BPs, notably proliferative bRG; (ii) a lengthening of the neurogenic period; (iii) an increased generation of neurons, in particular of upper-layer neurons; and (iv) an increase in the size of the CP that harbours these additional neurons, reflected by a greater thickness of its upper layers and a larger lateral length of the dorsal neocortex.

### Increase in proliferative bRG

The present effects of ARHGAP11B on BPs in developing ferret neocortex are substantially greater than, and significantly different from, those previously observed upon transient *ARHGAP11B* expression in embryonic mouse neocortex (*Florio et al., 2015*), with regard to BP quantity and quality. As to quantity, in embryonic mouse neocortex, *ARHGAP11B* expression resulted in a doubling and tripling of the pool size of total BPs and BPs in mitosis, respectively (*Florio et al., 2015*), whereas in embryonic ferret neocortex, *ARHGAP11B* expression led to nearly tripling and quintupling of the pool size of total BPs and BPs in mitosis (notably mitotic bRG), respectively (for a data summary, see *Supplementary file 1*). The greater effect of *ARHGAP11B* expression on mitotic BP abundance than total BP abundance, in both embryonic mouse and ferret neocortex, indicates that *ARHGAP11B* expression increases the proportion of the total duration of the BP cell cycle that is used for M-phase. This in turn would be consistent with ARHGAP11B promoting, in embryonic ferret neocortex, the proliferative rather than neurogenic mode of bRG division. This notion would be in line with the previous findings that both, cortical progenitors in embryonic mouse neocortex (*Arai et al., 2011*) and radial glial progenitors in postnatal ferret neocortex (*Turrero García et al., 2016*), which are not yet committed to neurogenesis spend a greater proportion of their total cell cycle in M-phase than those committed to neurogenesis.

Regarding the regulation of cell cycle length, it appears worthwhile to consider the present increase in BP abundance upon ARHGAP11B expression in developing ferret neocortex in the context of previous findings showing that shortening the duration of the G1 phase in progenitors of both mouse (*Lange et al., 2009*; *Pilaz et al., 2009*) and ferret (*Nonaka-Kinoshita et al., 2013*) developing neocortex results in increased BP proliferation and neocortex expansion. Furthermore, manipulation of cyclins or cdk/cyclin complexes was shown to lead to an increase in upper layer and callosal projection neurons (*Lange et al., 2009*; *Nonaka-Kinoshita et al., 2013*; *Pilaz et al., 2009*), which again is consistent with the phenotypes observed in the present study upon expression of ARHGAP11B in the developing ferret neocortex.

As to BP quality, mouse bRG have been shown to undergo mostly neurogenic cell divisions and to display a limited proliferative capacity (*Shitamukai et al., 2011*; *Wang et al., 2011*; *Wong et al., 2015*). In contrast, human bRG are known to be highly proliferative, which is thought to be one of the key factors contributing to the evolutionary expansion of the human neocortex (*Fietz et al., 2010*; *Florio and Huttner, 2014*; *Hansen et al., 2010*; *Lui et al., 2011*). The proliferative capacity of bRG in other mammalian species that have been studied appears to follow the general rule that highly proliferative bRG are more often found in species with an expanded neocortex (*Betizeau et al., 2013*; *Reillo et al., 2011*). Hence, the ARHGAP11B-promoted increase in the

number of proliferative (Sox2+ Tbr2–) bRG in the developing ferret neocortex fulfills a key requirement for further neocortex expansion.

In this context, it should be emphasized that those of the human-specific genes (*Dennis and Eichler, 2016*) that have been shown to be preferentially expressed in cortical progenitor cells (*Florio et al., 2018*) and that have been ectopically expressed in embryonic mouse neocortex, that is *ARHGAP11B* (*Florio et al., 2015*) and *NOTCH2NL* (*Fiddes et al., 2018*; *Florio et al., 2018*; *Suzuki et al., 2018*), show an increase in BP proliferation that mostly involves basal intermediate progenitors. To the best of our knowledge, our study provides first evidence of a human-specific gene increasing the number of proliferative bRG in a gyrencephalic neocortex. As bRG are considered to be instrumental for the evolutionary expansion of neocortex, this suggests that the ferret likely is a model system superior to mouse for studying the role of human-specific genes in neocortex development.

## Increase in upper-layer neurons

An increase in the upper layers is a fundamental characteristic of neocortex expansion (*Fame et al., 2011*; *Hutsler et al., 2005*; *Molnár et al., 2006*; *Tarabykin et al., 2001*). Layers II-IV are evolutionarily the most novel and mammalian-specific layers (*Molnár et al., 2006*). Layers II and III, in particular, underwent disproportional expansion during primate evolution. As the layers II-III neurons are the last-born neurons, an increase in the length of the neurogenic period might result in an increase in the number of late-born neurons. In fact, it has been proposed that the length of the neurogenic period may be a contributing factor in explaining the evolutionary increase in neocortex size and neuron number between humans and other great apes (*Lewitus et al., 2014*). Consistent with this, we observed a prolonged generation of late-born neurons, increased thickness of layers II-IV, and a marked increase in Satb2+ (3.3-fold) and, in particular, Brn2+ (5.6-fold) neurons in the upper layers of the CP, upon *ARHGAP11B* expression (for a data summary, see *Supplementary file 1*). Thus, also with regard to the key product generated by cortical progenitor cells, that is the neurons, ARHGAP11B is sufficient to promote another feature in the developing ferret neocortex that would be consistent with further neocortex expansion.

In this context, Satb2+ neurons are of special interest, as Satb2 is essential for establishing callosal projections to the contralateral hemisphere (*Alcamo et al., 2008*; *Britanova et al., 2008*). Callosal projection neurons are considered to play a key role in the high-level associative connectivity, thus contributing significantly to human cognitive abilities, with their impairment causing cognition-related pathologies (*Fame et al., 2011*). Given that the evolutionary increase in neocortex size is accompanied by an increase in callosal projection neurons, our data show that ARHGAP11B elicits yet another, specific feature in the developing ferret neocortex associated with further neocortex expansion.

## Neocortex expansion vs. increased thickness and lateral area

A hallmark of neocortex expansion is neocortical folding (*Borrell, 2018*; *Borrell and Götz, 2014*; *Kroenke and Bayly, 2018*). Transient expression of *ARHGAP11B* in the embryonic mouse neocortex induced cortical folding in about half of the cases in this normally lissencephalic rodent (*Florio et al., 2015*). However, expression of ARHGAP11B in the developing ferret neocortex did not significantly increase the GI, gyrus size, or sulcus depth and thickness in this gyrencephalic carnivore. The major morphological signs of neocortical expansion upon *ARHGAP11B* expression in developing ferret neocortex that we could detect were (i) an increased gyrus thickness, which reflected the already discussed increase in upper-layer thickness due to the increase in upper-layer neurons, and (ii) a larger lateral area of the CP where these neurons were found. The latter phenotype resulted in a ≈10% increase in the lateral length of the dorsal neocortex, which however did not appear to be sufficient to significantly increase the gyrus size or GI.

The reason why a 3–5–fold increase in upper-layer neurons (*Supplementary file 1*) resulted in more modest increases in upper-layer thickness and dorsal neocortex lateral length (24% and 10%, respectively) could be that either (i) the increase in the number of FP+ neurons occurred at the expense of FP– neurons, or (ii) that the FP+ neurons added themselves to the FP– neurons already existing in the control condition, thereby increasing neuronal cell density in the upper layers. Our results show that the latter scenario is the case, as we detected a significant increase in both the

total cell density and neuronal density in the upper cortical layers upon ARHGAP11B expression. Analysis of cortical neuron density as a function of cortical neuron number across carnivores has revealed that neuron density tends to decrease with increasing neuron number (*Herculano-Houzel et al., 2015*; *Jardim-Messeder et al., 2017*; *Lewitus et al., 2012*). This trend is much less prominent in primates, which in general exhibit a greater cortical neuron density per cortical neuron number than carnivores. Thus, the increase in upper-layer neuron density upon increasing upper-layer neuron number by the human-specific *ARHGAP11B* can be regarded as a form of 'primatization' of the ferret neocortex.

In this context, it is interesting to note that neocortical neuron density across carnivores shows a much greater range of variation than that across primates (*Herculano-Houzel et al., 2015*; *Lewitus et al., 2012*). We therefore hypothesize that the ferret neocortex is plastic enough to accommodate additional neurons, without having to increase neocortex size or folding. A corollary of this notion with regard to ARHGAP11B is that its primary function is to increase the proliferation and thus pool size of BPs, notably of bRG, which results in a lengthening of the neurogenic period and an increased generation of neurons, in particular of upper-layer neurons. In embryonic mouse neocortex, the ARHGAP11B-elicited increase in BPs could induce cortical folding as an indicator of neocortex expansion (*Florio et al., 2015*). In contrast, in the developing ferret neocortex studied here, the ARHGAP11B-elicited BP increase did not further increase the GI, although it did increase the neurogenic period and upper-layer neurons. This in turn resulted in a greater thickness of the upper layers, increased neuron density, and a larger lateral length of the dorsal neocortex. The difference between the phenotypes observed in mouse and ferret developing neocortex therefore appears to reflect how species in different mammalian orders tend to deal with an increase in cortical neuron number, rather than the primary function of ARHGAP11B.

# Materials and methods

**Key resources table**

| Reagent type (species) or resource | Designation | Source or reference | Identifiers | Additional information |
|---|---|---|---|---|
| Antibody | Goat polyclonal anti-Sox2 | R + D Systems | AF2018, RRID:AB_355110 | (1:200) |
| Antibody | Goat polyclonal anti-Sox2 | Santa Cruz Biotechnology | sc-17320 RRID:AB_2286684 | (1:200) |
| Antibody | Rabbit polyclonal anti-Tbr2 | Abcam | ab23345, RRID:AB_778267 | (1:250) |
| Antibody | Mouse monoclonal anti-PCNA | Millipore | CBL407, RRID:AB_93501 | (1:200) |
| Antibody | Mouse monoclonal anti-PhVim | Abcam | ab22651, RRID:AB_447222 | (1:200) |
| Antibody | Mouse monoclonal anti-Satb2 | Abcam | ab51502, RRID:AB_882455 | (1:200) |
| Antibody | Rabbit polyclonal anti-Tbr1 | Abcam | ab31940, RRID:AB_2200219 | (1:200) |
| Antibody | Rabbit polyclonal anti-Brn2 | Proteintech | 18998–1-AP, RRID:AB_10597389 | (1:100) |
| Antibody | Rabbit polyclonal anti-Olig2 | Millipore | ab9610, RRID:AB_570666 | (1:500) |

*Continued on next page*

*Continued*

| Reagent type (species) or resource | Designation | Source or reference | Identifiers | Additional information |
|---|---|---|---|---|
| Antibody | Rabbit polyclonal anti-GFAP | Dako | Z0334, RRID:AB_10013382 | (1:1000) |
| Antibody | Rabbit polyclonal anti-S100ß | Abcam | ab868, RRID:AB_306716 | (1:200) |
| Antibody | Rabbit polyclonal anti-active Caspase 3 | Abcam | ab2302 RRID:AB_302962 | (1:200) |
| Antibody | Mouse monoclonal anti-ARHG AP11B | MPI-CBG | | (1:100) |
| Antibody | Rabbit polyclonal anti-Arhgap11A | Abcam | ab113261, RRID:AB_10866587 | (1:500) |
| Antibody | Chicken polyclonal anti-GFP | Aves labs | GFP1020, RRID:AB_10000240 | (1:1000) |
| Antibody | Goat polyclonal anti-GFP | MPI-CBG | | (1:1000) |
| Antibody | Rat monoclonal anti-RFP | ChromoTek | 5F8, RRID:AB_2336064 | (1:500) |
| Antibody | Rabbit polyclonal anti-RFP | Rockland antibodies | 600-401-379, RRID:AB_2209751 | (1:1000) |
| Antibody | Goat polyclonal anti-Chicken Alexa Fluor 488 | ThermoFisher Scientific | A11039, RRID:AB_142924 | (1:500) |
| Antibody | Donkey polyclonal anti-Goat Alexa Fluor 488 | ThermoFisher Scientific | A11055, RRID:AB_2534102 | (1:500) |
| Antibody | Donkey polyclonal anti-Goat Alexa Fluor 555 | ThermoFisher Scientific | A21432, RRID:AB_2535853 | (1:500) |
| Antibody | Donkey polyclonal anti-Goat Alexa Fluor 647 | ThermoFisher Scientific | A21447, RRID:AB_141844 | (1:500) |
| Antibody | Donkey polyclonal anti-Rabbit Alexa Fluor 488 | ThermoFisher Scientific | A21206, RRID:AB_141708 | (1:500) |
| Antibody | Donkey polyclonal anti-Rabbit Alexa Fluor 555 | ThermoFisher Scientific | A31572, RRID:AB_162543 | (1:500) |
| Antibody | Donkey polyclonal anti-Rabbit Alexa Fluor 647 | ThermoFisher Scientific | A-31573, RRID:AB_2536183 | (1:500) or (1:1000) |
| Antibody | Donkey polyclonal anti-Mouse Alexa Fluor 488 | ThermoFisher Scientific | A-21202, RRID:AB_141607 | (1:500) or ('1:1000) |

*Continued on next page*

*Continued*

| Reagent type (species) or resource | Designation | Source or reference | Identifiers | Additional information |
|---|---|---|---|---|
| Antibody | Donkey polyclonal anti-Mouse Alexa Fluor 555 | ThermoFisher Scientific | A31570, RRID:AB_2536180 | (1:500) |
| Antibody | Donkey polyclonal anti-Mouse Alexa Fluor 647 | ThermoFisher Scientific | A31571, RRID:AB_162542 | (1:500) |
| Antibody | Goat polyclonal anti-Rat Alexa Fluor 555 | ThermoFisher Scientific | A21434, RRID:AB_141733 | (1:500) or (1:1000) |
| Antibody | Goat polyclonal anti-Rabbit Cy2 | Jackson Immuno research | # 111-225-144, RRID:AB_2338021 | (1:500) |
| Antibody | Goat polyclonal anti-Mouse Cy3 | Jackson Immuno research | #115-165-166, RRID:AB_2338692 | (1:500) |
| Antibody | Goat polyclonal anti-Mouse Cy5 | Jackson Immuno research | #115-175-166, RRID:AB_2338714 | (1:500) |
| Commercial assay or kit | Maxi prep kit | Qiagen | Cat#12362 | |
| Commercial assay or kit | RNeasy FFPE RNA isolation kit | Qiagen | Cat# 73504 | |
| Commercial assay or kit | NeuroTrace™ 640/660 deep-red fluorescent Nissl stain | Molecular probes | Cat# N-21483, RRID:AB_2572212 | |
| Commercial assay or kit | In Situ Cell Death Detection Kit, TMR red (TUNEL) | Sigma-Aldrich | 12156792910 | |
| Commercial assay or kit | Annexin V Cy5 reagent | Biovision | 1013–200 | |
| Commercial assay or kit | Click-iT EdU Alexa Fluor 647 Imaging Kit | Invitrogen | C10340 | |
| Strain, strain background (*Mustela putorius furo*) | Ferret | Marshall Bioresources | | |
| Strain, strain background (*Mustela putorius furo*) | Ferret | Euroferret | | |
| Sequence-based reagent | Hprt1 FqHprt1_F TACGCTGAG GATTTGGAAAAG | This paper | | oligonucleotide |
| Sequence-based reagent | Hprt1 FqHprt1_R CCATCTCCTT CATCACGTCTC | This paper | | oligonucleotide |
| Sequence-based reagent | ARHGAP11B qARHGAP11B_1F CAGAAAAGA AGGGCGTGTAC | This paper | | oligonucleotide |

*Continued on next page*

*Continued*

| Reagent type (species) or resource | Designation | Source or reference | Identifiers | Additional information |
|---|---|---|---|---|
| Sequence-based reagent | ARHGAP11B qARHGAP11B_1R GGAGTAGCAC AGAGACCATCA | This paper | | oligonucleotide |
| Sequence-based reagent | ARHGAP11B qARHGAP11B_2F TGAGAATAAGA TGGATAGCAGCA | This paper | | oligonucleotide |
| Sequence-based reagent | ARHGAP11B qARHGAP11B_2R GGTACACGCC CTTCTTTTCTG | This paper | | oligonucleotide |
| Recombinant DNA reagent | pCAGGS | (*Florio et al., 2015*) | | |
| Recombinant DNA reagent | pCAGGS-ARHGAP11B | (*Florio et al., 2015*) | | |
| Recombinant DNA reagent | pCAGGS-mCherry | (*Tavano et al., 2018*) | | |
| Recombinant DNA reagent | pCAGGS-GFP | (*Fei et al., 2014*) | | |
| Software, algorithm | Fiji/ImageJ | Fiji/Imagej | https://imagej.nih.gov/ij/ | |
| Software, algorithm | Prism | GraphPad software | | |
| Software, algorithm | ZEN | Carl Zeiss | | |

## Experimental animals

All experimental procedures were conducted in agreement with the German Animal Welfare Legislation after approval by the Landesdirektion Sachsen (licences TVV 2/2015 and TVV 21/2017). Timed-pregnant ferrets (*Mustela putorius furo*) were obtained from Marshall BioResources (NY, USA) or Euroferret (Copenhagen, Denmark) and housed at the Biomedical Services Facility (BMS) of MPI-CBG. Observed mating date was set to E0. Animals were kept in standardized hygienic conditions with free access to food and water and with an 16 hr/8 hr light/dark cycle. All experiments were performed in the dorsolateral telencephalon of ferret embryos, at a medial position along the rostro-caudal axis, in the prospective motor and somatosensory cortex.

## Plasmids

All plasmids used in this study were previously published (see the Key Resources table). All plasmids were extracted and purified using the EndoFree Plasmid Maxi kit (QIAGEN) following the manufacturer's instructions.

## *In utero* electroporation of ferrets

*In utero* electroporation of ferrets was performed as originally established by Dr. Hiroshi Kawasaki with the modifications listed below (*Kawasaki et al., 2012*). Pregnant jills (with embryos at E33) were kept fasted for at least 3 hr before the surgery. They were first placed in the narcosis box with 4% isoflurane. When in deep anesthesia, the ferrets were placed on the operation table and attached to the narcosis mask with a constant 3% isoflurane flow. Subsequently, the ferrets were injected subcutaneously with analgesic (0.1 ml Metamizol, 50 mg/kg), antibiotic (0.13 ml Synulox, 20 mg/kg or 0.1 ml amoxicilin, 10 mg/kg) and glucose (10 ml 5% glucose solution). A drop of Dexpanthenol Ointment solution was placed on their eyes to prevent eye dehydration during the surgical procedure. The ferret bellies were then shaved, sterilized with iodide and surgically opened. The uterus was exposed and embryos were injected intraventricularly with a solution containing 0.1% Fast Green (Sigma) in sterile PBS, 2.5 µg/µl of either pCAGGS vector (control) or pCAGGS-

ARHGAP11B vector, in either case together with 1 µg/µl of pCAGGS vector encoding a fluorescent protein (pCAGGS-GFP or pCAGGS-mCherry). For embryonic studies where the position of the electroporated embryos in the uterus was known, the same fluorescent protein-encoding vector was co-electroporated together with either control or ARHGAP11B-expressing vectors. For postnatal studies, pCAGGS and pCAGGS-ARHGAP11B were co-electroporated with different fluorescent protein-encoding vectors to enable distinction of kits. For different experiments, these vectors were alternated. Electroporations were performed with six 50-msec pulses of 100 V at 1 s intervals. Following electroporation, uterus was placed back in the peritoneal cavity and the muscle layer with the peritoneum were sutured using a 4–0 suture. The skin was sutured intracutaneously using the same thickness of the suture. Animals were carefully monitored until they woke up and then underwent postoperative care for the following 3 days (2 x daily 10 mg/kg amoxicilin, 3 x daily 25 mg/kg Metamizol).

### EdU labeling of ferret kits

For the neurogenic length experiments, a single pulse of EdU was injected intraperitoneally at P5, that is 13 days after *in utero* electroporations, and the animals were sacrificed 11 days later, at P16. The protocol for EdU injection was the same as the one previously published for ferret kits (*Turrero García et al., 2016*).

### Isolation of ferret brains

Ferret embryos were isolated at E37 and E40. Ferret kits were sacrificed at P0, P10 and P16. For the isolation of embryos, pregnant jills underwent the second surgery that followed the same pre-operative care, anesthesia and analgesia as the first surgery. The sutures from the first operation were carefully removed and the uterus was exposed. A caesarian section was made and embryos were removed from the uterus. Subsequently a complete hysterectomy was performed, after which the muscle layer with peritoneum and skin were sutured and the animal underwent the same post-operative care as after the first surgery. Embryonic brains were isolated and fixed at 4°C for 48 hr in 4% paraformaldehyde in 120 mM phosphate buffer pH 7.4.

For the postnatal time points, the kits were sacrificed by intraperitonal injection of 4 mg/kg Xylazin +40 mg/kg Ketamin. When in deep anaesthesia, the kits were perfused intracardially with PBS, followed by perfusion with 4% paraformaldehyde in 120 mM phosphate buffer pH 7.4 at room temperature. Kit brains were then isolated and fixed for 48 hr in 4% paraformaldehyde in 120 mM phosphate buffer pH 7.4 at 4°C.

Ferret jills whose kits were used postnatally also underwent the second surgery with the hysterectomy. Upon this surgery, they underwent the same post-operative care as after the first surgery. All jills were kept at the BMS of the MPI-CBG for at least two weeks after the second surgery. Afterwards they were donated for adoption. No adult ferrets were sacrificed in this study.

### Preparation of ferret brain slices

Upon fixation ferret brains were sectioned either on a vibratome or cryostat. Vibratome sections were 70 µm thick. Thickness of cryosections varied from 35 to 60 µm. Vibratome sections were either immunostained freshly prepared or conserved in cryoprotectant solution (30% sucrose, 30% ethylene glycol, 1% PVP40, 1.3 mM $NaH_2PO_4$, 3.9 mM $Na_2HPO_4$, 15 mM NaCl) and stored at –20°C for later use.

### Gene expression analysis by RT-qPCR

For analysis of *ARHGAP11B* expression, total RNA was isolated from two to three cryosections per developmental stage, using the RNeasy FFPE RNA isolation kit (Qiagen) including DNase-treatment following the manufacturer's instructions. An additional DNase-treatment was performed on the isolated RNA using the DNA-free DNA Removal Kit (Life technologies). cDNA was synthesized using random hexamers and Superscript III Reverse Transcriptase (Life Technologies). qPCR was performed using the Light Cycler SYBR green Master mix (Roche) on a Light Cycler 96 (Roche). Gene expression data were normalized based on the housekeeping gene *Hprt1*. Primer sequences are provided in the Key Resources Table.

## Immunofluorescence

Immunofluorescence was performed as previously described (*Kalebic et al., 2016*). Antigen retrieval (1 hr incubation with 10 mM citrate buffer pH 6.0 at 70°C in a water bath or oven) was performed for the following samples: all E37 samples, all E40/P0 samples, and vibratome sections of P10 and P16 samples which were used for immunostainings of the nuclear markers (Satb2, Brn2, Sox2, Olig2 and Ki67). When immunofluorescence for Tbr2 (E40/P0) was performed, a modified antigen retrieval protocol was used (1 hr incubation with 10 mM citrate buffer pH 6.0 supplemented with 0.05% Tween-20 at 80°C in a water bath). Antigen retrieval was followed by three washes with PBS.

All the immunostainings, except the one for ARHGAP11B, were done as follows. Samples were subjected to permeabilization for 30 min in 0.3% Triton X-100 in PBS at room temperature, followed by quenching for 30 min in 0.1 M glycine in PBS at room temperature. Blocking was performed in a blocking solution (0.2% gelatin, 300 mM NaCl, 0.3% Triton X-100 in PBS) for 30 min. Primary antibodies were incubated in the blocking solution for 48 hr at 4°C. Subsequently, the sections were washed three times in the blocking solution, incubated with secondary antibodies (1:500) and DAPI (Sigma) in the blocking solution for 1 hr at room temperature, and washed again three times in the blocking solution before being either used for Nissl post-staining or directly mounted on microscopy slides with Mowiol.

For the immunostaining for ARHGAP11B, permeabilization was performed for 1 hr in the 0.3% Triton X-100 solution at room temperature. Quenching was performed as for the other immunostainings. Blocking was done in 10% horse serum supplemented with 0.3% Triton X-100 (HS blocking solution) for 1 hr. Primary antibodies were incubated in the same solution for 24 hr, at 4°C. Subsequently, sections were washed five times in the HS blocking solution, incubated with the secondary antibodies (1:1000) and DAPI in the HS blocking solution for 1 hr at room temperature, washed five times in PBS and mounted on microscopy slides with Mowiol. ARHGAP11B antibody used in this study is a mouse monoclonal antibody raised against a recombinant full length ARHGAP11B. As the first 220 amino acids of ARHGAP11B are 89% identical to the first 220 amino acids of the ferret Arhgap11a, we showed by immunostaining that the antibody we used did not recognize the endogenous ferret Arhgap11a (*Figure 1—figure supplement 1E*).

Staining with the Nissl fluorescent stain, EdU detection, annexin V labeling, and TUNEL labeling were performed after the antibody stainings. NeuroTrace[TM] 640/660 deep-red fluorescent Nissl stain (Molecular probes) was used, following the manufacturer's instructions. EdU staining was performed using Click-iT EdU Alexa Fluor 647 Imaging kit (Invitrogen), following the manufacturer's instructions. Annexin V labeling was performed using the Annexin V-Cy5 bright fluorescence reagent (BioVision), following the manufacturer's instructions. TUNEL labeling was performed using the In situ cell death detection kit, TMR red (Sigma-Aldrich), following the manufacturer's instructions.

## Image acquisition

Images of whole ferret brains (*Figure 4—figure supplement 3A*) were obtained as follows. Images were acquired using an iPhone 7 and taking a photography of the brain through a UV-protection filter mounted on a SZX 16 Olympus stereomicroscope, equipped with a fluorescence lamp.

Fluorescent images were acquired using a Zeiss LSM 880 upright single-photon point scanning confocal system. For the embryonic stages, images were taken as either 1 μm single optical sections with the 40x objective or 2 μm single optical sections with the 20x objective. For the postnatal stages, images were taken as either 2 μm single optical sections with the 20x objective or 7.2 μm single optical sections with the 10x objective. When the images were taken as tile scans, the stitching of the tiles was performed using the ZEN software. Subsequently, all images were analyzed and processed with ImageJ (http://imagej.nih.gov/ij/).

## Quantifications

All cell counts were performed in standardized microscopic fields using Fiji, processed using Excel (Microsoft), and results were plotted using Prism (GraphPad Software). For each condition, data (typically at least three microscopic fields) from one experiment (see definition below) were pooled, and the mean of the indicated number of experiments was calculated. Whenever possible the quantifications were done blindly.

The definition of the morphological parameters is depicted graphically in the *Figure 4—figure supplement 3D*. All the morphological parameters except the lateral length of the dorsal neocortex were quantified at two positions (referred to as positions 1 and 2) in the somatosensory cortex, as defined in *Figure 4—figure supplement 3B*, for the following gyri and sulci: posterior sigmoid gyrus, coronal gyrus, lateral gyrus, cruciate sulcus, suprasylvian sulcus and lateral sulcus (*Sawada and Watanabe, 2012*). All the morphological parameters are presented as a ratio of the value of the electroporated hemisphere and the value of the contralateral hemisphere, as established previously (*Matsumoto et al., 2017*). Calculations were as follows.

Local Gyrification Index (Local GI): $\frac{The\ length\ of\ the\ inner\ contour}{The\ length\ of\ the\ outer\ contour}$

Local gyrification index is calculated as a ratio of the local GI of the electroporated area and the local GI of the equivalent area on the contralateral side. Gyrus size is calculated as a ratio of the size of an electroporated gyrus ad the size of the equivalent gyrus of the contralateral side. Sulcus depth is calculated as a ratio of the length of the line connecting the outer contour and the bottom of a sulcus and the equivalent line on the contralateral side. Sulcus thickness is calculated as a ratio of the thickness measured from the ventricular surface to the bottom of an electroporated sulcus and the equivalent thickness on the contralateral side. Gyrus thickness is calculated as a ratio of the thickness measured from the ventricular surface to the top of an electroporated gyrus and the equivalent thickness on the contralateral side. The upper layers thickness was measured as distance between the top of layer II and bottom of layer IV.

Lateral length of the FP+ area was measured as the distance between the medial-most and the lateral-most FP+ cell on a coronal cross-section, following the inner contour of the neocortex. The lateral length of the dorsal neocortex was measured at three positions along the rostro-caudal axis (positions 1 and 2, and position 1.5 located between positions 1 and 2 (*Figure 4—figure supplement 5C* bottom)), and was defined as a distance between the cingulate gyrus and ectosylvian gyrus along the inner contour, as depicted in *Figure 4—figure supplement 5A* bottom.

Total cell density was measured as number of DAPI+ nuclei in 50 μm x 50 μm fields. Neuronal density was measured as number of Satb2+ nuclei in the 50 μm x 50 μm fields.

## Statistical analysis

All statistics analyses were conducted using Prism (GraphPad Software). Sample sizes are reported in each figure legend, where the term 'one experiment' would refer to one embryo for analysis at E37 or E40, and to one kit for postnatal analysis. Total number of litters analyzed was as follows: E37, two litters; E40/P0, four litters; P10, two litters; P16, three litters. Embryos or kits from all litters were included in the statistical analyses. Tests used were Two-way ANOVA with Bonferroni posttest and Student's *t*-test. For each quantification, the statistical test and significance are indicated in the figure legend.

## Acknowledgements

We are grateful to the Services and Facilities of the Max Planck Institute of Molecular Cell Biology and Genetics for the outstanding support provided, notably J Helppi and his team of the Biomedical services (BMS) for the excellent husbandry of ferrets, J Peychl and his team of the Light Microscopy Facility and P Keller and his team of the Antibody Facility. We would like to particularly thank Anke Münch-Wuttke for help with perfusion of ferret kits and Dr. Anna Pfeffer for exceptional veterinary support. NK was supported by an EMBO long-term fellowship (ALTF 861–2013). CG acknowledges support from the Erasmus+ traineeship program and MA from the Christiane-Nüsslein-Volhard Foundation. WBH was supported by grants from the DFG (SFB 655, A2), the ERC (250197) and ERA-NET NEURON (MicroKin).

## Additional information

### Funding

| Funder | Grant reference number | Author |
|--------|------------------------|--------|
| European Molecular Biology Organization | ALTF 861-2013 | Nereo Kalebic |
| Deutsche Forschungsgemeinschaft | SFB 655, A2 | Wieland B Huttner |
| European Research Council | 250197 | Wieland B Huttner |
| Max-Planck-Gesellschaft | | Wieland B Huttner |
| Christiane-Nüsslein-Volhard Foundation | | Mareike Albert |
| Erasmus+ | Traineeship program | Carlotta Gilardi |

The funders had no role in study design, data collection and interpretation, or the decision to submit the work for publication.

### Author contributions

Nereo Kalebic, Conceptualization, Formal analysis, Investigation, Visualization, Methodology, Writing—original draft, Writing—review and editing, Performed ferret in utero electroporations; Carlotta Gilardi, Investigation, Visualization, Methodology, Assisted with ferret surgeries; Mareike Albert, Investigation, RNA isolation, RT-qPCR and gene expression analysis; Takashi Namba, Resources, ARHGAP11B construct and antibody; Katherine R Long, Milos Kostic, Methodology, Assisted with ferret surgeries; Barbara Langen, Methodology, Ferret hysterectomy, Assisted with ferret in utero electroporations; Wieland B Huttner, Conceptualization, Supervision, Funding acquisition, Project administration, Writing—review and editing

### Author ORCIDs

Nereo Kalebic (iD) http://orcid.org/0000-0002-8445-2906
Mareike Albert (iD) http://orcid.org/0000-0001-9855-9344
Katherine R Long (iD) http://orcid.org/0000-0003-0660-2486
Wieland B Huttner (iD) http://orcid.org/0000-0003-4143-7201

### Ethics

Animal experimentation: All experimental procedures were conducted in agreement with the German Animal Welfare Legislation after approval by the Landesdirektion Sachsen (licences TVV 2/2015 and TVV 21/2017).

### Decision letter and Author response

Decision letter https://doi.org/10.7554/eLife.41241.021
Author response https://doi.org/10.7554/eLife.41241.022

## Additional files

### Supplementary files

• Supplementary file 1. Overview of the effects of ARHGAP11B expression on neural progenitor cells and upper-layer neurons in developing ferret neocortex. Data taken from the indicated figure panels were used for the calculations shown. For the calculations pertaining to upper-layer neurons, it is assumed that the ARHGAP11B-induced increase in the lateral length of the FP+ area in the rostro-caudal dimension is equal to that in the medio-lateral dimension (1.25-fold).
DOI: https://doi.org/10.7554/eLife.41241.018

• Transparent reporting form
DOI: https://doi.org/10.7554/eLife.41241.019

## Data availability

All data generated or analysed during this study are included in the manuscript and supporting files.

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
