## [Decision Letter]

Thank you for submitting your article "Human-specific ARHGAP11B induces hallmarks of neocortical expansion in developing ferret neocortex" for consideration by *eLife*. Your article has been reviewed by three peer reviewers and the evaluation has been overseen by a Reviewing Editor and Marianne Bronner as the Senior Editor. The following individual involved in review of your submission has agreed to reveal their identity: Victor Borrell (Reviewer #2).

The reviewers have discussed the reviews with one another and the Reviewing Editor has drafted this decision to help you prepare a revised submission.

Summary:

This is an important study that clearly adds to the discussion around gyrification mechanisms and the genetic drivers of such a critical event of brain development in humans. There is agreement among the reviewers that the work is very important, the experiments well executed and the manuscript well written. The finding that expression of ARHGAP11B in developing ferret cortex leads to increased proliferation of basal progenitors, increased neurogenesis in upper layers, and increased tangential dispersion of neurons but not to an increase in cortex folding, is extremely interesting because this is very different from what seen in mice, which have a smooth cortex.

From our collective analysis, revisions are minimal should concentrate on some main points.

Essential revisions:

First, a more detailed analysis of tangential migration of neurons is necessary. Because of variability in the location of electroporated cells in each animal, it becomes difficult to compare across experiments. The context of the electroporation may be different and thus neurons may migrate differently simply because the environment is distinct (and tangential migration varies by rostro-caudal location). One way to do this would be to co-electroporate the control and experimental construct in the same animal, with distinct reporters.

Second, neuronal density cannot be estimated by measuring fluorescence. More accurate counts are needed.

Third, the thickness of the SVZ needs to be measured in the ARGHAP11B condition (Figure 1) to make sure that there is not discrepancy between thickness of SVZ and the fact that PCNA+ cells did not increase.

Fourth, some text modifications are required, especially in the Introduction and Discussion to integrate the current data within a framework of other published work that looked at the relationship between proliferation of basal progenitors and gyrification in distinct species. Details are outlined below, in the detailed reviews. Reviewer #2 highlights specific regions of the manuscript where more clarity and discussion of prior data is needed.

*Reviewer #2:*

This is a fascinating and excellently executed study that aims at understanding the molecular and cellular mechanisms underlying the evolutionary expansion of the human cerebral cortex. This lab discovered some years ago a gene present uniquely in the human genome (ARHGAP11B) and highly expressed in human cortical progenitor cells, and they found that expression of this human-specific gene in the developing mouse cortex caused very significant increases in mouse cortical progenitor proliferation and, in turn, it induced folding of the otherwise smooth mouse cortex. In this current study, the authors express this same gene in the developing cerebral cortex of ferret, which naturally develops a folded cortex, aiming to identify any different or additive effects on cortex development with respect to their previous analyses in mouse. The authors conclude that expression of ARHGAP11B in developing ferret cortex leads to increased abundance of proliferative basal progenitors, increased neurogenesis in upper layers, and increased tangential dispersion of cortical neurons (all typical of primates), but enigmatically not to increased cortex folding. The study is well conducted, the manuscript is very well written, figures of high quality and the results are clearly presented. A clear example of a manuscript that should be published in *eLife*.

My main concern with this study is that some of the main conclusions are based on results that have serious limitations due to the very nature of the experimental design. In particular, effects on the tangential dispersion of cortical neurons cannot be determined by using in utero electroporation, because each different electroporation in a different animal targets a different extent of VZ, and so the tangential extent of cortex containing labeled neurons is also inherently different, and thus not at all comparable between animals nor treatments. If tangential dispersion of neurons is increased upon ARHGAP11B expression, is cortical surface area affected? One would expect to maybe have folding increased, but this is also not. One possible explanation is that non-electroporated cells compensate for those expressing ARHGAP11B, by undergoing less tangential dispersion than normal. It may clarify this important issue (including faithful comparison of experimental conditions) if the authors could do the simple experiment of labeling non-ARHGAP11B cells in the same area containing the ARHGAP11B-expressing cells.

A similar problem is related to neuron density. The density of fluorescent neurons in the cortex completely depends on the density of electroporated cortical progenitor cells, which varies significantly from animal to animal, even between control animals. Therefore, in this case again, solid conclusions cannot be drawn about neuron density in absolute terms. Because these are two key points in the authors' major, they represent major concerns for me on this, otherwise fantastic, manuscript.

*Reviewer #3:*

The submitted manuscript addresses a very exciting topic and presents a straightforward, well conducted, series of results regarding the effects of overexpressing a human-specific gene specifically expressed in the basal progenitors of the cerebral cortex in the ferret cortex. Interestingly, this work shows that ARGHAP11B over expression in the ferret cortex yields significantly different results from that obtained in the lissencephalic mouse. Indeed, in their seminal study (Florio et al., 2015), the authors reported a de novo formation of sulci in mouse brain at E18, following electroporation at E15. In the ferret, ARGHAP11B over expression has a massive effect on the OSVZ with a 3.5 fold increase in the proportion of PCNA+ progenitors. When performed at E33, ARGHAP 11B leads to a higher proportion of neurons forming layers 2, 3 and 4 being generated, accompanied by an increase in the density of the layers 3 and 4 of the cerebral cortex without significantly modifying the gyrification pattern-albeit an increase of thickness of the affected gyri. The lateral and media lateral dimensions of the electroporated regions are also increased by 25%.

The novel results by Kalebic et al. inform on species-specific constraints regarding cortical expansion mechanisms. In this context, the present results should be discussed with respect to previous results from the same group (Lange et al., 2009) and others (Pilaz et al., 2009) showing an increase in cortical surface and an increase in the number and density of layers 2 and 3 cortical neurons respectively following an increase of the basal pool of progenitors in the mouse as well as with respect to the results of Nonata-Kinoshita et al., 2013, reporting differential effects of forced amplification of basal progenitors in the mouse (no effect on gyrification) and in the ferret (increased gyrification and increase in cortical surface).

---

## [Author Response]

Essential revisions:First, a more detailed analysis of tangential migration of neurons is necessary. Because of variability in the location of electroporated cells in each animal, it becomes difficult to compare across experiments. The context of the electroporation may be different and thus neurons may migrate differently simply because the environment is distinct (and tangential migration varies by rostro-caudal location). One way to do this would be to co-electroporate the control and experimental construct in the same animal, with distinct reporters.

We have now performed a more detailed analysis of the tangential migration of neurons and its consequences for the lateral expansion of the neocortex. We have now included 6 new panels (in Figure 4 and newly-added Figure 4—figure supplement 5) regarding this analysis, as is described below.

We have not, however, performed the experiment to co-electroporate the control and experimental construct in the same animal, with distinct reporters, as the suggested experimental design would result in the vast majority of cells co-expressing both control and ARHGAP11B constructs, rendering the interpretation of data very difficult. Additionally, this experiment would have required new electroporations at E33 and raising ferrets up to P16, i.e. a total of almost 2 months before we would have been able to begin analyzing the samples; this would have made it impossible to meet the 2-month deadline for resubmission.

1) To account for potential differences in the rostro-caudal location, we have now measured the lateral length of the area harboring fluorescent protein-expressing cells (FP+ area) at 3 different positions along the rostro-caudal axis. This revealed that there are no differences in the lateral length of the FP+ area along the rostro-caudal axis for control and ARHGAP11B expression, respectively, with a significantly higher lateral length of the FP+ area upon ARHGAP11B expression compared to control in each of the three positions (Figure 4—figure supplement 5C).

2) Next, to account for the effects of the electroporation itself, we have now measured the lateral length of the FP+ area at an earlier stage, E40/P0 (Figure 4—figure supplement 5D, E). This showed that at this earlier stage there are no differences in apical and basal length of the FP+ area between ARHGAP11B expression and control.

3) Finally, we have examined the effects of the tangential migration on the lateral expansion of the neocortex. We have measured the lateral length of the dorsal neocortex (as depicted in Figure 4—figure supplement 5A bottom) and, importantly, found a 10% greater lateral length upon ARHGAP11B expression (Figure 4H). Of note, also this increase was observed at each of the three different positions along the rostro-caudal axis (Figure 4—figure supplement 5F). This new data imply that other, fluorescent protein-negative (FP–) cells do not compensate the effects of ARHGAP11B expression and that they do not undergo less tangential migration as a consequence, a possibility raised by reviewer 2 below.

The lack of compensation is also consistent with data presented in the original version of our manuscript (now Figure 4—figure supplement 5G, H), showing that expression of ARHGAP11B caused an increase in total cell density in the upper layers of the CP (please see also our response to the next comment in the Editors' Decision Letter).

Second, neuronal density cannot be estimated by measuring fluorescence. More accurate counts are needed.

We believe that there is a misunderstanding here. Neuronal density was not determined by counting fluorescent protein-positive cells (i.e. progeny of electroporated progenitors) in specific layers of the CP, but by counting DAPI-stained nuclei per area, irrespective of whether a nucleus was in a fluorescent protein-positive or -negative cell (see revised Figure 4—figure supplement 5H). This is quite accurate, and we have clarified this in the revised manuscript.

Furthermore, we have added another quantification of neuronal density that, again, is independent of fluorescent protein expression. Specifically, we have now quantified the density of the Satb2+ nuclei in the upper layers of the CP. As was the case for total cell density (Figure 4—figure supplement 5H), we have found an increase in Satb2+ neuron density in layers III and IV upon ARHGAP11B expression (see revised Figure 4—figure supplement 5I).

Third, the thickness of the SVZ needs to be measured in the ARGHAP11B condition (Figure 1) to make sure that there is not discrepancy between thickness of SVZ and the fact that PCNA+ cells did not increase.

We have now measured the thickness of the VZ, ISVZ and OSVZ and have not detected any difference between ARHGAP11B-expressing and control neocortex (Figure 1—figure supplement 2F).

As to the issue of the PCNA+ cells in the SVZ, our previous data (now revised Figure 1B) showed that the *proportion* of PCNA+ FP+ cells was increased in the OSVZ, but not ISVZ, upon ARHGAP11B expression. This may have led to a misunderstanding. To clarify, we now additionally present the *abundance* of PCNA+ FP+ cells in the ISVZ and OSVZ (Figure 1—figure supplement 2B). This analysis reveals that the total number of FP+ PCNA^+^ increased in both ISVZ and OSVZ, but the magnitude of this increase is far greater in the OSVZ. This is consistent with our previous data that the proportion of FP+ cells in the OSVZ that are PCNA+ increased (revised Figure 1B).

Fourth, some text modifications are required, especially in the Introduction and Discussion to integrate the current data within a framework of other published work that looked at the relationship between proliferation of basal progenitors and gyrification in distinct species. Details are outlined below, in the detailed reviews. Reviewer #2 highlights specific regions of the manuscript where more clarity and discussion of prior data is needed.

We have revised the text as requested and as specified in our response to the reviewers' comments below.

Reviewer #2:[…] My main concern with this study is that some of the main conclusions are based on results that have serious limitations due to the very nature of the experimental design. In particular, effects on the tangential dispersion of cortical neurons cannot be determined by using in utero electroporation, because each different electroporation in a different animal targets a different extent of VZ, and so the tangential extent of cortex containing labeled neurons is also inherently different, and thus not at all comparable between animals nor treatments. If tangential dispersion of neurons is increased upon ARHGAP11B expression, is cortical surface area affected? One would expect to maybe have folding increased, but this is also not. One possible explanation is that non-electroporated cells compensate for those expressing ARHGAP11B, by undergoing less tangential dispersion than normal. It may clarify this important issue (including faithful comparison of experimental conditions) if the authors could do the simple experiment of labeling non-ARHGAP11B cells in the same area containing the ARHGAP11B-expressing cells.

We have now performed a more detailed analysis of the tangential migration of neurons and its consequences for the lateral expansion of the neocortex. We have now included 6 new panels (in Figure 4 and newly-added Figure 4—figure supplement 5) regarding this analysis, as is described below.

To account for potential differences in the rostro-caudal location, we have now measured the lateral length of the area harboring fluorescent protein-expressing cells (FP+ area) at 3 different positions along the rostro-caudal axis. This revealed that there are no differences in the lateral length of the FP+ area along the rostro-caudal axis for control and ARHGAP11B expression, respectively, with a significantly higher lateral length of the FP+ area upon ARHGAP11B expression compared to control in each of the three positions (Figure 4—figure supplement 5C).

Next, to account for the effects of the electroporation itself, we have now measured the lateral length of the FP+ area at an earlier stage, E40/P0 (Figure 4—figure supplement 5D, E). This showed that at this earlier stage there are no differences in apical and basal length of the FP+ area between ARHGAP11B expression and control.

As to the issue of a potential increase in cortical surface area upon ARHGAP11B expression:

We have now examined the effects of the tangential migration on the lateral expansion of the neocortex. We have measured the lateral length of the dorsal neocortex (as depicted in Figure 4—figure supplement 5A bottom) and, importantly, found a 10% greater lateral length upon ARHGAP11B expression (Figure 4H). Of note, also this increase was observed at each of the three different positions along the rostro-caudal axis (Figure 4—figure supplement 5F). This new data imply that other, fluorescent protein-negative (FP–) cells do not compensate the effects of ARHGAP11B expression and that they do not undergo less tangential migration as a consequence, a possibility raised by the reviewer.

The lack of compensation is also consistent with data presented in the original version of our manuscript (now Figure 4—figure supplement 5G, H), showing that expression of ARHGAP11B caused an increase in total cell density in the upper layers of the CP.

We have not, however, performed the experiment of labeling non-ARHGAP11B cells in the same area containing the ARHGAP11B-expressing cells suggested by the Reviewer. Such an experiment would have required new electroporations at E33 and raising ferrets up to P16, i.e. a total of almost 2 months before we would have been able to begin analyzing the samples; this would have made it impossible to meet the 2-month deadline for resubmission.

A similar problem is related to neuron density. The density of fluorescent neurons in the cortex completely depends on the density of electroporated cortical progenitor cells, which varies significantly from animal to animal, even between control animals. Therefore, in this case again, solid conclusions cannot be drawn about neuron density in absolute terms. Because these are two key points in the authors' major, they represent major concerns for me on this, otherwise fantastic, manuscript.

We believe that there is a misunderstanding here. Neuronal density was not determined by counting fluorescent protein-positive cells (i.e. progeny of electroporated progenitors) in specific layers of the CP, but by counting DAPI-stained nuclei per area, irrespective of whether a nucleus was in a fluorescent protein-positive or -negative cell (see revised Figure 4—figure supplement 5H). This is quite accurate, and we have clarified this in the revised manuscript.

Furthermore, we have added another quantification of neuronal density that, again, is independent of fluorescent protein expression. Specifically, we have now quantified the density of the Satb2+ nuclei in the upper layers of the CP. As was the case for total cell density (Figure 4—figure supplement 5H), we have found an increase in Satb2+ neuron density in layers III and IV upon ARHGAP11B expression (see revised Figure 4—figure supplement 5I).

Reviewer #3:[…] The novel results by Kalebic et al. inform on species-specific constraints regarding cortical expansion mechanisms. In this context, the present results should be discussed with respect to previous results from the same group (Lange et al., 2009) and others (Pilaz et al., 2009) showing an increase in cortical surface and an increase in the number and density of layers 2 and 3 cortical neurons respectively following an increase of the basal pool of progenitors in the mouse as well as with respect to the results of Nonata-Kinoshita et al., 2013, reporting differential effects of forced amplification of basal progenitors in the mouse (no effect on gyrification) and in the ferret (increased gyrification and increase in cortical surface).

We thank the reviewer for this constructive and insightful suggestion, which we have followed. Specifically, we now discuss our findings with respect to (i) previous studies that manipulated the duration of the G1 phase of the cortical progenitor cell cycle and (ii) the consequences of this manipulation on BP proliferation, neocortex expansion and upper-layer neurons (subsection “Increase in proliferative bRG”).